# Lignin: An Adaptable Biodegradable Polymer Used in Different Formulation Processes

**DOI:** 10.3390/ph17101406

**Published:** 2024-10-21

**Authors:** Andreea Creteanu, Claudiu N. Lungu, Mirela Lungu

**Affiliations:** 1Department of Pharmaceutical Technology, University of Medicine and Pharmacy Grigore T Popa, 700115 Iași, Romania; acreteanu@gmail.com; 2Department of Functional and Morphological Science, Faculty of Medicine and Pharmacy, Dunarea de Jos University, 800010 Galati, Romania; mirelacrainiciuc@gmail.com

**Keywords:** LIG, polymers, drug molecules, biomolecules, biosynthesis, antimicrobial properties

## Abstract

Introduction: LIG is a biopolymer found in vascular plant cell walls that is created by networks of hydroxylated and methoxylated phenylpropane that are randomly crosslinked. Plant cell walls contain LIG, a biopolymer with significant potential for usage in modern industrial and pharmaceutical applications. It is a renewable raw resource. The plant is mechanically protected by this substance, which may increase its durability. Because it has antibacterial and antioxidant qualities, LIG also shields plants from biological and chemical challenges from the outside world. Researchers have done a great deal of work to create new materials and substances based on LIG. Numerous applications, including those involving antibacterial agents, antioxidant additives, UV protection agents, hydrogel-forming molecules, nanoparticles, and solid dosage forms, have been made with this biopolymer. Methods: For this review, a consistent literature screening using the Pubmed database from 2019–2024 has been performed. Results: The results showed that there is an increase in interest in lignin as an adaptable biomolecule. The most recent studies are focused on the biosynthesis and antimicrobial properties of lignin-derived molecules. Also, the use of lignin in conjunction with nanostructures is actively explored. Conclusions: Overall, lignin is a versatile molecule with multiple uses in industry and medical science

## 1. Introduction

Lignin (LIG), a complex organic polymer, is a critical player in plant biology. Its primary building blocks, monolignols, are linked through various chemical bonds, resulting in a heterogeneous and irregular structure. This unique structure gives LIG several crucial functions. It provides mechanical strength and rigidity to plant cell walls. It assists in transporting water and nutrients through the vascular tissues by strengthening the xylem vessels. It contributes to the plant’s defense against pathogens and pests due to its complex and recalcitrant structure, which is difficult for most organisms to break down. These functions underscore the importance of LIG in plant biology and biochemistry [1].

Furthermore, the global lignin market was valued at USD 1.08 billion in 2023 and is projected to expand at a compound annual growth rate (CAGR) of 4.5% from 2024 to 2030. The rising demand for lignin in animal feed and natural goods is expected to stimulate growth. The product is extensively employed in the synthesis of macromolecules for the manufacture of bitumen, biofuels, and biorefinery catalysts. This aspect is expected to facilitate market expansion. The COVID-19 pandemic adversely affected the market. This occurred due to the closure of manufacturing facilities and plants as a result of the lockdown and limitations. Disruptions in supply chains and transportation exacerbated obstacles for the sector. The industry encountered a backlash due to disturbances in the value chain, encompassing staff reductions, raw material shortages, trade and transportation challenges, and unpredictable customer demand [2].

LIG’s industrial and environmental significance is vast. As a byproduct of the paper-making process, it is used to produce high-quality paper. It is also a valuable source of energy when burned and finds use in various industrial applications. Moreover, as a renewable resource, LIG is being explored for conversion into biofuels, chemicals, and materials, contributing to the development of sustainable biorefineries. In natural ecosystems, the degradation of LIG plays a significant role in the carbon cycle, influencing soil fertility and carbon sequestration. These applications underscore the potential of LIG in various industries and environmental contexts [3].

Applications of LIG are vast. LIG can be used in the production of adhesives, resins, and binders for various industrial applications. Research is ongoing to develop LIG-based bioplastics as a sustainable alternative to petroleum-based plastics. LIG is being investigated as a precursor for the production of carbon fiber, which is used in lightweight, high-strength materials for the aerospace and automotive industries.

The heterogeneous and complex nature of LIG makes it challenging to process and utilize efficiently. Significant research is focused on developing methods to break down LIG into valuable monomers and chemicals through processes such as pyrolysis, oxidation, and microbial degradation [4].

LIG’s unique properties and abundance make it a promising material for various industrial applications, although its complexity presents challenges that continue to be the subject of active research and innovation (Figure 1).

### 1.1. LIG Isolation

A sustainable society must look for renewable resources to replace fossil carbon in the production of chemicals, fuels, and materials. A significant portion of lignocellulosic biomass, LIG, is a plentiful and renewable source of aromatics that is currently underutilized since it is frequently burned as an unwanted byproduct in the manufacture of bioethanol and paper. There is much promise for this LIG as a source of valuable aromatic compounds and materials. LIG-focused biorefinery methods are presently being developed with the goal of obtaining additional value from LIG. Nonetheless, the degree of degradation and modification of the extracted LIG directly affects how well these innovative LIG-focused biorefineries function. Hence, in order to potentially enhance the worth of LIG, it is imperative to thoroughly investigate the reactivity and degradation routes of the inherent LIG present in the plant material. The structure of undegraded native-like LIG derived from lignocellulosic plant material closely resembles that of native LIG. This renders it a good candidate for a comprehensive investigation into the reactivity and structure of native LIG [5].

One of the main problems in converting lignocellulose biomass into products with additional value is LIG recalcitrance. Although the use of LIGolytic bacteria is currently restricted, in situ biodegradable LIG-modifying enzyme-producing bacteria are thought to be an excellent answer to LIG biodegradation issues [6].

As an illustration, researchers examined the potential of LIG, a substance extracted from black liquor waste, for the production of binderless briquettes composed entirely of LIG. These briquettes had a calorific value that varied between 5670 and 5876 kcal/kg. The acid precipitation technique was employed to extract LIG from black liquor. Throughout the precipitation process, the pH level was controlled at different levels by using sulfuric acid, citric acid, and acetic acid. The evaluation of isolation conditions was conducted utilizing a range of approaches, such as the Klasson method, proximal analysis, ultimate analysis, Fourier transform infrared (FTIR), adiabatic bomb calorimetry, density measurement, and drop shatter index (DSI) testing. The methodologies employed aimed to examine the influence of isolation circumstances on the attributes of LIG and the qualities of the resultant briquette. The results demonstrated that briquettes with a fixed carbon content of 72%, an exceptional degree of sulfonation index of 99.7%, and a calorific value comparable to coal-based briquettes were effectively produced from LIG extracted using citric acid at a pH of 3 [7].

Acidifying the pH of the mixture enhances the extraction of LIG from black liquor. Sulfuric acid is particularly effective for LIG precipitation due to its capacity to lower the pH of the solution to a minimum of 2 [8]. A lower pH is required for the separation of LIG from black liquor and for the protonation of its phenolic and carboxylic functional groups. However, the use of sulfuric acid has significant drawbacks, such as the release of hazardous gases that can lead to environmental problems and carcinogenic effects in humans [9,10]. Employing a less potent acid is considered to be a viable solution for these problems. The structure of lignin, cellulose, and hemicellulose is shown in Figure 2.

### 1.2. LIG Synthesis

LIG valorization can serve as a substitute for certain aromatic compounds as a platform due to its composition as a polymer of aromatic molecules. Some bacteria possess methods for degrading LIG. So far, researchers have discovered more and more ways to convert LIG-derived aromatic compounds. This makes it simpler to create more pathways for LIG conversion in model microorganisms. The utilization of synthetic biology techniques would promote the conversion of LIG into valuable products such as aromatic compounds, ring-cleaved chemicals produced from LIG, and bioactive substances [11].

Recent findings have shown that LIG has the potential to produce a range of aromatic chemicals, such as gallic acid, vanillin, and p-hydroxybenzoic acid (pHBA). pHBA, an industrial platform chemical, is commonly used as the initial raw material for the biosynthesis of many commercially essential compounds such as vanillyl alcohol, xiamenmycin, gastrodin, resveratrol, and MA. Additionally, it is utilized as a monomer in the production of liquid crystal polymers. The Burkholderia glumae strain BGR achieved effective production of pHBA from the LIG component p-coumaric acid by removing two genes involved in pHBA degradation. In addition, it was shown that increasing the expression of phcs II, which encodes for p-hydroxcinnmaoyl-CoA synthetase II, improved the pHBA synthesis in batch reaction. In addition, when *E. coli* expresses a distinct pHBA decarboxylase, it has the ability to convert pHBA into phenol, which is a commonly utilized chemical in the chemical industry [12].

Vanillin is an essential aromatic flavoring component used in innovative food and perfume fragrances. The removal of vanillin dehydrogenase in *Rhodococcus jostii* RHA1 (*R. jostii* RHA1) led to the accumulation of vanillin, reaching a concentration of 96 mg/L. This was seen in a minimum medium containing 2.5% wheat straw lignocellulose and 0.05% glucose. The process involved the use of a temperature-controlled system in *E. coli* to convert ferulic acid, a typical component of LIG, into vanillin. This was achieved by expressing Fcs and Ech enzymes from the thermophilic actinomycete *A. thermoflava* N1165. At a temperature of 50 °C, vanillin was produced in this system as a result of the decrease in the activities of ADHs and the increase in the activities of functional Fcs and Ech at high temperatures. Vanillin was synthesized at a temperature of 30 °C in this particular system. However, the constant requirement for ATP and CoA has greatly hindered the effective transformation of LIG-derived aromatics due to the biosynthetic pathway involving Fcs and Ech. In addition, Pad and Ado established a pathway in *E. coli* that does not require a coenzyme to produce 4-vinylguaiacol and vanillin from ferulic acid. Vanillin was synthesized in *S. cerevisiae* by converting the LIG component ferulic acid using the VpVAN gene from Vanilla planifolia through heterologous expression. ADH, an alcohol dehydrogenase in *S. cerevisiae*, efficiently catalyzes the transformation of vanillin into vanillyl alcohol [13].

In addition, the detoxification study revealed that *S. cerevisiae* was capable of converting p-coumaric acid, ferulic acid, and coniferyl aldehyde into metabolites that are less toxic. Aminated compounds are very valued synthetic materials that play a crucial role in the creation of a wide range of exceptional chemicals and pharmaceuticals. Through the process of heterologous production, the w-transaminase (w-Tam) known as CV2025 from *Chromobacterium violaceum* DSM30191 may be synthesized in *E. coli*. This synthesis enables the conversion of vanillin into vanillylamine through amination. The enzyme vanillin dehydrogenase, encoded by the genes vdh or ligV, catalyzes the conversion of vanillin to vanillic acid, except for vanillylamine. Vanillic acid exhibits pharmacological effects. Aryl O-demethylase possesses the capacity to convert vanillic acid into protocatechuic acid (PCA), which serves as an essential constituent of polymers. The production of PCA from vanillic acid in *E. coli* was enhanced by introducing plant methionine synthase and vanillic acid O-demethylase (ligM) from SYK-6. The VanA and VanB genes found in LIG-degrading bacteria, such as *R. jostii* RHA and Pseudomonas putida KT2440, are also responsible for catalyzing the O-demethylation of vanillic acid. In addition, Corynebacterium glutamicum can produce p-coumaric acid (PCA) from p-hydroxybenzoic acid (pHBA) through the expression of the pobA gene, which encodes the pHBA hydroxylase. Catechol serves as a valuable precursor for the production of diverse polymeric polymers, and it can be obtained through the decarboxylation of PCA. In order to decrease the harmfulness, vanillin was converted into catechol in *E. coli* by a synthetic process involving the genes ligV, ligM, and aroY (which encode PCA decarboxylase) and resulting in the production of an aromatic transporter called CouP. In addition, the vanillin self-inducible promoter ADH7 from *S. cerevisiae* was introduced into E. coli. to circumvent the cost of using an external inducer. Furthermore, the expression of a three-domain reductase (GcoB) and a cytochrome P450 aromatic O-demethylase from the CYP255A family in *P. putida* can facilitate the conversion of guaiacol, a consequence of LIG depolymerization, into catechol [14].

Syringaldehyde, a LIG-derived aromatic molecule, can be transformed into syringic acid through the heterologous expression of desV or ligV in *E. coli*. In addition, the demethylase enzymes desA and ligM have the ability to convert syringic acid into gallic acid, which is a naturally occurring phytochemical known for its strong antibacterial and antioxidant characteristics. In addition, gallic acid was produced in *E. coli* by using pHBA, with the expression of the Pseudomonas aeruginosa pHBA hydroxylase mutant PobA in a heterologous manner. The lpdc gene catalyzes the decarboxylation of gallic acid, resulting in the production of pyrogallol. Pyrogallol serves as a precursor for the manufacture of trimethoprim, an antibiotic, muscle relaxant gallamine triethiodide, and insecticide bendiocarb [9,12,14,15]. Syringaldehyde structure is represented in Figure 3.

Examples of LIG extraction are listed in the table below (Table 1):

## 2. Production of Bioactive Substances

Flavonoids, stilbenoids, and coumarins are known to be frequently utilized in human health care as compounds produced from phenylpropanoid. The LIG pretreatment liquid contained p-coumaric acid and ferulic acid, which were utilized to create artificial biosynthetic pathways in *E. coli* or *S. cerevisiae*, resulting in the production of flavonoids, stilbenoids, and coumarins Flavonoids, stilbenoids, and coumarins are commonly employed in human healthcare as chemicals derived from phenylpropanoid. The LIG pretreatment liquid contains p-coumaric acid and ferulic acid, which were used to construct synthetic pathways in *E. coli* or *S. cerevisiae*, leading to the synthesis of flavonoids, stilbenoids, and coumarins. In addition, resveratrol can be synthesized from pHBA by engineering a reverse β-oxidative phenylpropanoid degradation pathway in Corynebacterium glutamicum strains. The presence of a sugar residue significantly alters the bioactivity of the majority of secondary metabolites, which are predominantly found in their glycosylated form in nature. Gastrophin was synthesized in *E. coli* by constructing a synthetic pathway that involved endogenous alcohol dehydrogenases, Rhodiola glycosyltransferase UGT73B6, and Nocardia carboxylic acid reductase. This process allowed the conversion of pHBA into dystrophin. Vanillin glucoside was produced from ferulic acid by simultaneously expressing VpScVAN and AtUGT72E2 in *S. cerevisiae*. Overexpression of the VvGT2 gene from Vitis vinifera in *E. coli* resulted in the glycosylation of gallic acid, a consequence of LIG breakdown, leading to the production of β-glucogallin. Based on these discoveries, it is possible to create artificial routes to transform aromatic molecules originating from LIG, such as p-coumaric acid, ferulic acid, and pHBA, into bioactive substances that have increased worth, such as flavonoids, stilbenoids, and coumarins, along with their glycosylated derivatives. Furthermore, by developing a reversal of the β-oxidative phenylpropanoid degradation pathway in Corynebacterium glutamicum strains, resveratrol can be obtained from pHBA. A sugar residue dramatically changes the bioactivity of most secondary metabolites, which are found in nature mostly in their glycosylated form. By creating an artificial pathway including endogenous alcohol dehydrogenases, Rhodiola glycosyltransferase UGT73B6, and Nocardia carboxylic acid reductase, gastrophin was generated from pHBA in *E. coli*. By coexpressing VpScVAN and AtUGT72E2 in *S. cerevisiae*, vanillin glucoside was generated from ferulic acid. When the glucosyltransferase gene VvGT2 from Vitis vinifera was overexpressed in *E. coli*, gallic acid—a byproduct of the breakdown of LIG—was glycosylated, yielding β-glucogallin. According to these findings, synthetic pathways might be built to convert LIG-derived aromatics like p-coumaric acid, ferulic acid, and pHBA into bioactive compounds with added value like flavonoids, stilbenoids, and coumarins, as well as their glycosylated derivatives [32,33]

## 3. LIG Bioactivity: Antimicrobial and Antifungal Properties

LIG bioactivity pertains to the physiological and biochemical effects of LIG and its derivatives on living creatures, encompassing plants, animals, and microbes. The many bioactivities of these substances can be applied in the fields of medicine, pharmacy, agriculture, and environmental management. The following are key elements of LIG bioactivity: LIG and its derivatives have been shown to possess antibacterial properties against various bacterial strains. This makes LIG a potential natural preservative or antimicrobial agent in food and medical and pharmaceutical applications. Also, LIG compounds can inhibit the growth of certain fungi, offering potential for use in agricultural settings to protect crops from fungal infections. LIG can disrupt bacterial cell walls and membranes, leading to cell lysis and death. Its phenolic components can also interfere with bacterial enzymes and proteins, inhibiting their function (Table 2).

These references cover a broader range of mechanisms through which LIG exhibits antimicrobial activity, including membrane potential disruption, interaction with membrane proteins, inhibition of ATP synthesis, induction of apoptosis-like cell death, activation of antimicrobial peptides, synergistic effects with other antimicrobials, and alteration of microbial metabolic pathways. LIG is effective against a wide range of bacteria, including both Gram-positive and Gram-negative strains. This broad-spectrum activity makes it a versatile antimicrobial agent. When used in combination, LIG can enhance the efficacy of traditional antibiotics. This synergistic effect can help reduce the required antibiotic doses and mitigate the development of antibiotic resistance [11].

## 4. Research and Application

Modification for Enhanced Activity: Chemical modifications of LIG, such as sulfonation or oxidation, can enhance its antimicrobial properties. Modified LIG often exhibits more substantial and more specific antimicrobial effects. Combination with Other Agents: LIG can be combined with traditional antimicrobial agents to enhance efficacy and reduce the likelihood of resistance development. Incorporating LIG into nanomaterials can improve its delivery and interaction with microbial cells, making it more effective as an antimicrobial agent in coatings, films, and other applications [51].

LIG’s antimicrobial activity can vary depending on its source, extraction method, and degree of modification. Standardizing LIG preparations is necessary for consistent results. While LIG is generally considered safe, its potential toxicity to human cells and the environment needs to be thoroughly assessed, especially for applications in medicine and food preservation (Table 3).

### 4.1. LIG as a Binder

Furthermore, briquette applications have been described as using LIG as a binder [59,60,61,62,63]. LIG was used as a binder in earlier studies for wood [64,65], organic municipal solid waste [66], and anthracite-based briquettes [67,68]. Additionally, it was stated that LIG might be used as a fuel and binder, enhancing the briquette’s thermal and physical characteristics. Densified particles or loose fuel components make up briquettes. In briquetting, loose fuel components or particles are crushed under pressure into smaller volume agglomerates that can hold their compressed state [69]. It has certain advantages to using biomass directly as fuel, including a higher heat content and a more compact size that makes storage and transportation easier [70]. Typically, coal or biomass, such as agricultural waste, is used to make briquettes; binder is sporadically used as well. When a material is unable to generate a robust densified form, a binder is required. Its function is to promote inter-particulate bonding, which enables the densified form to form [71,72]. Even though the inorganic-based binder creates a briquette with a greater compressive strength and compaction ratio and is more hydrophobic, the organic-based binder is favored [73]. This is due to the fact that adding inorganic materials tends to lower the calorific value of the final briquette and raise its ash content and burn-out temperature [74,75]. Because of the differing thermal behaviors of the binder and fuel material, binder-based briquettes, although flexible, have several drawbacks, including non-uniform combustion qualities and a loss in compacting properties at high temperatures [76,77]. In these situations, using the binderless briquette—a briquette compacted without binder—is more beneficial. However, the materials that may be utilized in this kind of briquette are restricted since the materials must be able to create a strong attraction between all of the particles. It should be possible to condense LIG, the material often employed as the binder, into an all-LIG, binderless briquette. Its important qualities as fuel and the capacity of LIG molecules to form hydrogen bonds with one another—a feature made possible by the hydroxy group (-OH) in LIG—made this conceivable. This study used the acid precipitation method to separate and purify LIG from black liquor, which was then used to create an all-LIG briquette without the need for a binder. Citric and acetic acids, which are less harmful to the environment than sulfuric acid, were used to isolate LIG. It has been shown that LIG can precipitate in the presence of acetic and citric acids [78]. The study assessed the impact of pH and acid type on the LIG purity and briquette characteristics. To the best of our knowledge, LIG—which is primarily used as a binder—has never been used in a single-component binderless briquette. Density measurement, proximal and ultimate analysis, the calorific value test, and the drop shatter test were used to evaluate the briquette’s properties [79,80,81].

LIG is a highly prevalent organic polymer on Earth, ranking second in abundance only to cellulose. LIG is mainly located in the cell walls of plants and provides stiffness, defense, and the ability to transport water. It plays a crucial function in maintaining the structural strength of plants. Due to its intricate and diverse chemical composition, which is composed of phenolic subunits, this molecule is very adaptable and can be used in a wide range of industries. LIG’s versatility is being utilized more and more in fields such as material science, energy generation, pharmaceuticals, cosmetics, and environmental sustainability, making it an essential component in the advancement of bio-based economies [82].

### 4.2. Structural Adaptability in Nature; Role in Plants

LIG is crucial for the mechanical strength of plant cell walls, particularly in woody plants. It cross-links with other cell wall components such as cellulose and hemicellulose, creating a robust matrix that allows plants to grow tall and withstand environmental stresses like wind and microbial attack. The hydrophobic nature of LIG also facilitates efficient water transport within the plant, particularly in vascular tissues like the xylem, where it helps prevent water loss and supports nutrient distribution [83].

LIG is primarily composed of three types of phenylpropanoid monomers: p-coumaryl alcohol, coniferyl alcohol, and sinapyl alcohol. These monomers form different kinds of LIG depending on their relative abundance. Guaiacyl (G) LIG: Predominantly composed of coniferyl alcohol, typically found in gymnosperms (softwoods). Syringyl (S) LIG: Predominantly composed of sinapyl alcohol, commonly found in angiosperms (hardwoods). p-Hydroxyphenyl (H) LIG: Contains more p-coumaryl alcohol, often found in grasses and some other plants [84].

The structure of LIG can vary significantly based on the type and ratio of monomers and the types of linkages (such as β-O-4, β-β, and β-5 linkages) that form between them. This results in a highly heterogeneous and irregular polymer network, which is adaptable to different functional requirements. LIG provides rigidity and compressive strength to plant cell walls, allowing plants to grow tall and withstand various physical stresses. LIG’s structure contributes to the water-repellent nature of plant cell walls, preventing water loss and providing protection against pathogens. The complex and irregular structure of LIG makes it resistant to enzymatic breakdown, contributing to its role in plant defense [85].

LIG’s structure can be influenced by environmental factors such as light, temperature, and nutrient availability. For example, plants exposed to high levels of UV radiation may produce more LIG with certain monomer compositions that provide better protection against UV damage. Stress conditions, such as drought or pathogen attack, can also lead to alterations in LIG biosynthesis, resulting in a more robust or differently cross-linked LIG structure to protect the plant better. The structural adaptability of LIG has been crucial in plant evolution, particularly in the development of vascular plants that can grow tall and transport water efficiently. This adaptability has allowed plants to colonize a wide range of terrestrial environments. LIG’s ability to evolve and adapt its structure has also played a role in the co-evolution of plant–microbe interactions, particularly in the context of LIG degradation by certain fungi and bacteria [86].

### 4.3. Biodegradation Resistance

The intricate and uneven composition of LIG renders it remarkably impervious to microbial decomposition. This attribute enhances the resilience of wood and other lignified tissues in the natural environment, facilitating the long-term retention of carbon in terrestrial ecosystems. The inherent durability of LIG against decomposition presents difficulties in the processing of biomass for biofuel production, necessitating the use of sophisticated techniques to convert LIG into usable energy efficiently. The adaptability of LIG has significant implications for industries such as biofuel production, where modifying LIG content and structure in plants could improve the efficiency of biomass conversion. Understanding LIG’s structural adaptability also aids in developing new materials, such as bioplastics and carbon fibers, where LIG’s natural properties can be harnessed or modified for specific applications. LIG’s structural adaptability is a key factor in its biological functions, its role in plant evolution, and its potential for various industrial applications [87].

### 4.4. Industrial Versatility: Material Science

LIG’s chemical versatility makes it a valuable component in the production of bioplastics and composites. It can be chemically modified and blended with other polymers, enhancing the material’s properties, such as biodegradability, strength, and thermal stability. These LIG-based materials are increasingly used in packaging, automotive components, construction materials, and more, offering sustainable alternatives to petroleum-based plastics. LIG can be integrated into thermoplastic matrices to enhance their mechanical properties and biodegradability. LIG-based composites exhibit improved stiffness, UV resistance, and thermal stability, which are essential for automotive components, electronics, and packaging materials. Moreover, these materials contribute to reducing the environmental footprint associated with conventional plastics [88].

LIG can be used as a renewable raw material to produce bioplastics. It can be chemically modified and blended with other biopolymers like polylactic acid (PLA) or polyhydroxyalkanoates (PHA) to create sustainable alternatives to conventional plastics. LIG is often used as a filler or reinforcing agent in composite materials, improving mechanical properties such as stiffness and strength. For example, LIG can be combined with polymers like polypropylene or polyethylene to create biocomposites that are lighter and more environmentally friendly than traditional composites. LIG is a promising precursor for carbon fiber production due to its aromatic structure and high carbon content. LIG-derived carbon fibers are being developed as a cost-effective and sustainable alternative to those made from polyacrylonitrile (PAN). These fibers are used in high-performance materials for the aerospace, automotive, and sporting goods industries. LIG-based carbon fibers can also be used in energy storage applications, such as in the electrodes of batteries and supercapacitors, where they offer high conductivity and surface area. LIG can be used as a sustainable alternative to petroleum-based adhesives in wood products like plywood and particleboard. LIG-based adhesives provide strong bonding properties while being environmentally friendly and reducing formaldehyde emissions. LIG can be used as a binder in construction materials, such as asphalt and concrete, improving durability and reducing the environmental impact of these materials. LIG’s natural UV-absorbing properties make it an excellent additive for UV stabilization in plastics, coatings, and paints. It can also act as an antioxidant, protecting materials from oxidative degradation and extending their lifespan. LIG has inherent flame-retardant properties due to its high carbon content and char formation during combustion. It can be used as an additive in polymers to enhance flame resistance. LIG can be processed into nanoparticles, which have applications in drug delivery, coatings, and as functional fillers in polymers. These nanoparticles offer advantages such as biodegradability, low toxicity, and the ability to be functionalized for specific applications. LIG can be used to create hydrogels with tunable properties for applications in biomedical devices, water purification, and agriculture. These hydrogels benefit from LIG’s biodegradability and ability to form stable cross-linked networks. LIG can be depolymerized to produce valuable aromatic chemicals like vanillin, phenols, and other compounds used in the production of resins, adhesives, and various fine chemicals. This valorization of LIG aligns with the principles of green chemistry and circular economy. LIG-derived solvents, such as bio-based dimethyl sulfoxide (DMSO) alternatives, are being explored for their potential to replace traditional, petroleum-based solvents in various industrial processes. LIG can be used in water treatment applications as an adsorbent to remove heavy metals, dyes, and other pollutants from wastewater. Its functional groups allow it to bind with various contaminants, making it effective in environmental remediation. LIG can be used to improve soil structure and fertility, acting as a natural binder in soil stabilization processes or as a slow-release fertilizer when combined with other nutrients [89].

One of the challenges in using LIG in material science is the variability in its structure and composition depending on its source and extraction process. Research is ongoing to develop methods for purifying and modifying LIG to achieve consistent and desirable properties for various applications. While LIG has shown great potential in numerous applications, scaling up the production of LIG-based materials to industrial levels remains a challenge. This includes optimizing extraction methods and integrating LIG processing into existing manufacturing systems [90,91].

### 4.5. Energy Production

LIG’s high carbon content and energy density make it a promising candidate for biofuel production. It can be converted into bio-oil, biodiesel, or syngas through processes such as pyrolysis, gasification, or hydrothermal liquefaction. LIG-derived biofuels are renewable and can help reduce dependence on fossil fuels, contributing to lower greenhouse gas emissions. Additionally, LIG can be co-fired with other biomass in power plants to produce heat and electricity, enhancing the energy output and sustainability of biomass energy systems. LIG can be used to create environmentally friendly adhesives and resins. These LIG-based adhesives are formaldehyde-free, reducing the health risks associated with traditional adhesive products. Applications include wood composites, coatings, and laminates, where LIG serves as a sustainable alternative to synthetic resins [92].

LIG plays an increasingly significant role in energy production, particularly as a source of bioenergy. Its high carbon content and abundance in biomass make it a promising candidate for various energy-related applications. Below are some key areas where LIG is utilized in energy production. LIG, being a significant component of lignocellulosic biomass (about 15–30%), is burned directly in biomass power plants to generate heat and electricity. Its high energy content, typically around 25–27 MJ/kg, makes it an efficient fuel for combustion processes. The direct combustion of LIG contributes to renewable energy generation, especially in integrated biorefineries where LIG is a byproduct of biofuel production. Pyrolysis is a thermal decomposition process where LIG is heated in the absence of oxygen, leading to the production of bio-oil, syngas, and biochar. Bio-oil can be further refined into biofuels like biodiesel, while syngas can be used to generate electricity or as a precursor for synthetic fuels. In gasification, LIG is converted into syngas (a mixture of carbon monoxide and hydrogen) through partial oxidation at high temperatures. These syngas can be used to produce electricity or converted into liquid fuels through the Fischer–Tropsch synthesis. HTL is a process where LIG is converted into bio-crude oil under high pressure and temperature in the presence of water. The resulting bio-crude can be upgraded into transportation fuels. HTL has the advantage of being able to process wet biomass, making it a versatile option for LIG-rich materials. LIG can be processed into solid fuels such as pellets and briquettes, which are used in heating systems and small-scale power generation. These solid fuels have a high energy density and can be used as a renewable alternative to coal or wood in boilers and furnaces. LIG can be used as a feedstock for hydrogen production through processes like catalytic reforming. This involves breaking down LIG’s complex structure into smaller molecules, which are then converted into hydrogen gas. LIG-derived hydrogen is a potential renewable energy source for fuel cells and other applications. LIG can also be involved in photocatalytic water splitting, where it acts as a sacrificial agent to produce hydrogen. This method leverages solar energy to split water into hydrogen and oxygen, with LIG providing electrons to facilitate the process. Microbial fuel cells (MFCs) can utilize LIG as a substrate for electricity generation. Certain microorganisms can degrade LIG and, in the process, transfer electrons to an electrode, generating electricity. This is a novel approach to converting LIG waste into bioelectricity, although it is still in the experimental stage. LIG can be used to produce carbon materials for energy storage devices, such as batteries and supercapacitors. For instance, LIG-derived activated carbon can serve as an electrode material in supercapacitors, offering high surface area and good conductivity. LIG can be chemically modified to produce materials with redox-active properties, making it suitable for use in flow batteries or as a binder in lithium-ion batteries. LIG’s natural polymer structure can be tailored to enhance energy storage. In integrated biorefineries, LIG is often a byproduct of processes like ethanol production. Instead of being considered trash, LIG can be utilized through energy generation techniques such as combustion, gasification, or conversion into biofuels. This enhances the overall effectiveness and long-term viability of the biorefinery process. An obstacle in utilizing LIG for energy generation is its intricate and fluctuating composition, which poses difficulties in effectively converting it into energy products. Continual research is being conducted to create catalysts and techniques that can more efficiently dismantle the structure of LIG. The economic feasibility of generating energy from LIG is a crucial factor to consider. Although LIG is readily available, the expense of transforming it into energy products needs to be competitive with alternative sources of renewable and non-renewable energy. Technological advancements and optimization of processes are essential for enhancing the economic viability of LIG-based energy production [93,94,95].

### 4.6. Chemical Production

LIG can be depolymerized to produce a range of aromatic compounds, such as vanillin (a flavoring agent) and phenols (used in resins and adhesives). These aromatic chemicals are traditionally derived from petrochemicals, but LIG offers a more sustainable and renewable source. The ability to produce high-value chemicals from LIG is a crucial area of research, driving innovation in green chemistry and industrial biotechnology. LIG, as a complex and abundant biopolymer, holds significant potential for the production of various chemicals. The aromatic nature of LIG, derived from its phenylpropanoid units, makes it a valuable resource for creating high-value chemicals, especially in the context of sustainable and bio-based chemical production. Below are some critical applications and processes where LIG is utilized in chemical production. LIG, as a complex and abundant biopolymer, holds significant potential for the production of various chemicals. The aromatic nature of LIG, derived from its phenylpropanoid units, makes it a valuable resource for creating high-value chemicals, especially in the context of sustainable and bio-based chemical production. Below are some critical applications and processes where LIG is utilized in chemical production [96].

LIG is one of the primary natural sources of vanillin, a widely used flavoring agent. Vanillin can be produced through the oxidative depolymerization of LIG, offering a renewable alternative to synthetic vanillin derived from petrochemicals. This process typically involves the oxidation of LIG to break down its complex structure into smaller aromatic molecules, including vanillin. LIG can be depolymerized to produce phenolic compounds such as phenol, catechol, guaiacol, and syringol. These compounds are essential in the production of phenolic resins, adhesives, and other industrial chemicals. Phenolic compounds from LIG can be obtained through thermal or chemical depolymerization methods like pyrolysis or alkaline oxidation. Through depolymerization processes such as hydrolysis, pyrolysis, or hydrogenolysis, LIG can be broken down into monomers and oligomers. These smaller molecules serve as platform chemicals that can be further transformed into a wide range of products, including bio-based plastics, solvents, and other chemical intermediates. These monomers, derived from the breakdown of LIG, are valuable for synthesizing bio-based polymers. They can also be used in the production of specialty chemicals and as precursors for the synthesis of more complex molecules. LIG can be used as a feedstock to produce bio-based polyurethane foams. These foams are used in insulation, packaging, and cushioning materials. The polyols derived from LIG are reacted with isocyanates to create polyurethane, offering a more sustainable alternative to conventional petrochemical-derived polyurethanes. LIG can replace phenol in the production of phenolic resins, which are used in adhesives, coatings, and molding compounds. These LIG-based resins are more environmentally friendly and contribute to the reduction of reliance on fossil fuels. LIG-derived compounds have antioxidant and UV-absorbing properties, making them suitable for use as additives in plastics, cosmetics, and pharmaceuticals [97].

These compounds can enhance the stability and shelf life of products by protecting them from oxidative degradation and UV damage. Certain LIG-derived chemicals exhibit bioactive properties, making them potential candidates for pharmaceutical applications. For example, LIG-derived oligomers and monomers have shown antimicrobial, anti-inflammatory, and anticancer activities in various studies. LIG can be chemically modified to produce bio-based solvents, such as dimethyl sulfoxide (DMSO) alternatives. These solvents are helpful in multiple industrial processes, including chemical synthesis and formulation, offering a greener alternative to traditional solvents. Depolymerized LIG can yield dimeric and oligomeric compounds that serve as green solvents in various applications. These solvents are characterized by their biodegradability and low toxicity, making them attractive for environmentally conscious chemical processes. LIG can be used as a filler or reinforcing agent in composite materials. It enhances the mechanical properties of composites, making them stronger and more durable. These LIG-enhanced materials are used in automotive parts, construction materials, and consumer goods. Epoxy resins derived from LIG are used in coatings, adhesives, and composite materials. LIG-based epoxies offer improved sustainability compared to traditional petrochemical-based resins, and they can be tailored for specific applications by modifying the LIG structure.

Various catalytic processes have been developed to depolymerize LIG into valuable chemicals. These include hydrogenolysis, oxidation, and reductive depolymerization, which break down LIG’s complex structure into more straightforward, more valuable chemical compounds. Catalysts used in these processes often include metals like palladium, nickel, and ruthenium, which help facilitate the breaking of LIG’s strong bonds. Enzymatic methods, using LIG-degrading enzymes such as laccases and peroxidases, can selectively break down LIG into specific monomers and oligomers. These biocatalytic approaches offer mild reaction conditions and high specificity, making them attractive for producing fine chemicals from LIG. One of the significant challenges in using LIG for chemical production is its complex and heterogeneous structure, which varies depending on the source and extraction method. This variability can make it challenging to achieve consistent chemical yields and product quality. The economic viability of LIG-based chemical production depends on improving processing technologies to increase efficiency and reduce costs. Research and development in this area focus on optimizing LIG extraction, depolymerization, and conversion processes to make them more commercially feasible [98,99].

### 4.7. Environmental and Agricultural Applications: Soil Enhancement

LIG-based products can be used as soil conditioners. When applied to agricultural soils, LIG helps to increase organic matter content, promoting healthier plant growth and reducing the need for chemical fertilizers. Its ability to bind soil particles also helps prevent erosion and improves soil stability. LIG’s ability to adsorb heavy metals and organic pollutants makes it useful in water treatment and environmental remediation. LIG-based materials, such as hydrogels and activated carbons, are effective in removing contaminants from wastewater, offering a sustainable solution for ecological management. LIG can enhance soil structure by promoting the formation of soil aggregates. These aggregates improve soil porosity and aeration, which are crucial for root growth and microbial activity. The binding properties of LIG help stabilize these aggregates, reducing soil erosion and compaction. LIG’s ability to hold water makes it beneficial for improving soil water retention. When added to soil, LIG can increase the soil’s capacity to retain moisture, which is particularly valuable in arid and semi-arid regions. This improved water retention can lead to better plant growth and reduced irrigation needs. LIG contributes to the increase in soil organic matter (SOM), which is essential for soil fertility. As a complex and recalcitrant organic compound, LIG decomposes slowly, providing a long-term source of carbon and nutrients for the soil. This slow decomposition helps maintain a steady supply of organic matter, supporting microbial communities and enhancing soil health. The incorporation of LIG into the soil can aid in carbon sequestration, helping to mitigate climate change. Due to its resistance to degradation, LIG can remain in the soil for extended periods, effectively storing carbon and reducing atmospheric CO_2_ levels. LIG can enhance soil fertility by improving nutrient retention. It has a high cation exchange capacity (CEC), which allows it to retain essential nutrients like calcium, magnesium, potassium, and ammonium. These nutrients are then slowly released to plants, improving nutrient availability and reducing the need for chemical fertilizers. LIG can bind with heavy metals in the soil, reducing their bioavailability and toxicity. This chelation process can help remediate contaminated soils and make them safer for agricultural use. LIG can play a significant role in preventing soil erosion. When used as a soil amendment or in mulching, LIG helps stabilize the soil surface, reducing the impact of wind and water erosion. This is especially important in areas prone to erosion, where LIG can protect the soil and maintain its productivity. LIG-rich materials, such as bark or straw, can be used as mulch to protect the soil surface. These materials help reduce evaporation, control weeds, and prevent soil erosion while also adding organic matter as they decompose [100].

LIG provides a habitat for soil microorganisms, particularly fungi, which play a crucial role in LIG degradation. The presence of LIG in the soil supports a diverse and active microbial community, which is essential for nutrient cycling, soil structure, and overall soil health. LIG can influence the decomposition rates of other organic materials in the soil. By interacting with enzymes produced by soil microorganisms, LIG can modulate the breakdown of plant residues, contributing to a balanced nutrient release over time. LIG can be used as a base for biodegradable soil amendments that improve soil quality without leaving harmful residues. These amendments can be designed to slowly release nutrients, improve soil structure, and enhance microbial activity, providing a sustainable alternative to synthetic soil conditioners. LIG-containing organic matter, such as compost or biochar, can enhance the soil’s natural resistance to pests and diseases. The physical barrier provided by LIG-rich mulch can reduce the spread of soil-borne pathogens, while the enhanced microbial activity in LIG-amended soils can suppress harmful organisms.

While it contributes to long-term soil stability and carbon sequestration, it may not provide immediate nutrient availability compared to other organic amendments. Blending LIG with other, faster-decomposing organic materials can help balance nutrient release. The effectiveness of LIG as a soil enhancer can depend on its source and the method of extraction. LIG from industrial processes, such as those in the pulp and paper industry, may contain impurities or be chemically modified, which could affect its performance in soil applications. Ensuring the purity and suitability of LIG for agricultural use is essential [101,102,103,104,105].

### 4.8. Biodegradable Mulches

LIG-based mulches are used in agriculture to suppress weeds, retain soil moisture, and regulate soil temperature. Unlike traditional plastic mulches, LIG-based mulches are biodegradable and contribute to soil health as they decompose, adding organic matter and improving soil fertility over time. This practice supports sustainable agriculture by reducing the need for herbicides and synthetic inputs. LIG can be used to create slow-release fertilizers, where nutrients are encapsulated within a LIG matrix. This allows for a more controlled release of nutrients, improving their efficiency and reducing environmental impacts such as nutrient runoff, which can cause water pollution. The slow decomposition rate of LIG can be both an advantage and a challenge. LIG-based biodegradable mulches are an emerging, eco-friendly alternative to conventional plastic mulches used in agriculture and landscaping. These mulches offer several environmental and agricultural benefits due to their biodegradability, contribution to soil health, and ability to suppress weeds. Here is an overview of LIG biodegradable mulches, their benefits, and how they are used [106,107].

LIG for biodegradable mulches is typically derived from industrial byproducts, such as those from the paper and pulp industry, or agricultural residues like straw or wood chips. The LIG is extracted and processed into a material that can be used as mulch. LIG can be combined with other natural polymers (e.g., cellulose, hemicellulose) and bioplastics to create a composite material that is durable yet biodegradable. These composites can be formed into sheets or films, similarly to traditional plastic mulches. The production of LIG-based mulches involves blending LIG with other biodegradable polymers, possibly adding fillers, plasticizers, and other additives to enhance performance. The material is then processed through extrusion or casting methods to create mulch films or sheets. Unlike traditional plastic mulches, LIG-based mulches break down naturally in the soil over time, reducing waste and environmental pollution. Microorganisms in the soil decompose the mulch, converting it into carbon dioxide, water, and biomass, which further enrich the soil. LIG mulches effectively suppress weeds by blocking sunlight, preventing weed seeds from germinating and growing. This reduces the need for chemical herbicides, promoting a more organic approach to weed management [108].

LIG mulches help retain soil moisture by reducing evaporation. This is particularly beneficial in arid and semi-arid regions, where water conservation is crucial. By maintaining higher soil moisture levels, these mulches can improve plant growth and reduce the frequency of irrigation. Similarly to plastic mulches, LIG-based mulches help regulate soil temperature by insulating the soil. They can keep the soil warmer in cool climates and cooler in hot climates, thus creating a more stable environment for plant roots. As LIG-based mulches decompose, they add organic matter to the soil, enhancing soil structure, fertility, and microbial activity. The slow release of LIG’s complex organic compounds can improve soil carbon content and nutrient availability over time. LIG-based biodegradable mulches offer a sustainable alternative to polyethylene (PE) and other conventional plastic mulches, which contribute to plastic pollution. By decomposing naturally, LIG mulches eliminate the need for removal and disposal, reducing labor costs and environmental impact. The use of LIG-based mulches can contribute to carbon sequestration. As LIG decomposes, it helps build soil organic matter, which acts as a long-term carbon sink, potentially mitigating climate change. Traditional plastic mulches can break down into microplastics, which pose risks to soil health, water quality, and wildlife. LIG-based mulches avoid this issue, as they fully biodegrade into non-toxic substances [109,110].

While biodegradability is a significant advantage, ensuring that the mulch lasts throughout the growing season without degrading too quickly can be a challenge. Researchers are working on optimizing the balance between durability and biodegradability. Currently, LIG-based biodegradable mulches may be more expensive than traditional plastic mulches due to the costs associated with production and the development of new materials. However, as production scales up and technology advances, costs are expected to decrease. The performance of LIG-based mulches can vary depending on the specific formulation and environmental conditions. Factors such as temperature, humidity, and soil microbial activity can influence the rate of degradation and the effectiveness of the mulch. LIG-based mulches are particularly well-suited for use in vegetable and fruit crops, where they can provide weed suppression, moisture retention, and soil temperature regulation. They are especially beneficial in organic farming systems, where synthetic chemicals are avoided. Beyond agriculture, LIG-based mulches can be used in landscaping for weed control, aesthetic purposes, and soil enhancement. Their natural appearance and environmental benefits make them an attractive option for sustainable landscaping practices. These mulches can also be used in row crops and perennial plantings, where long-term soil health is a priority. The gradual decomposition of LIG adds to the organic matter over time, improving soil quality for future plantings. Ongoing research focuses on optimizing the composition of LIG-based mulches to enhance their performance characteristics, such as strength, flexibility, and degradation rate. This involves experimenting with different LIG sources, blends with other biodegradable materials, and the incorporation of natural additives. Field trials are being conducted to assess the effectiveness of LIG-based mulches in various agricultural and environmental settings. These trials help determine the best practices for using these mulches and provide data on their long-term impact on soil health and crop yields [111,112].

### 4.9. Pharmaceutical and Cosmetic Uses: Drug Delivery Systems

LIG nanoparticles have emerged as promising carriers for drug delivery systems. These nanoparticles can encapsulate active pharmaceutical ingredients, allowing for targeted delivery and controlled release in the body. LIG’s biocompatibility and ability to bind with a wide range of molecules make it a versatile platform for developing new drug delivery technologies, particularly in cancer treatment and chronic disease management. LIG has inherent antioxidant properties due to its phenolic structure, which can neutralize free radicals and prevent oxidative damage. These properties make LIG a valuable ingredient in pharmaceuticals and cosmetics, particularly in products designed to protect the skin from aging. LIG also exhibits antimicrobial properties, which can be utilized in wound dressings and antimicrobial coatings [113].

Vanillin, namely 4-hydroxy-3-methoxybenzaldehyde, is typically found in the most significant amounts, making up approximately 20% of the total. It is currently the sole phenolic chemical produced on a large scale from biomass. The pod of the Vanilla orchid accounts for only 5% of global vanilla production. By contrast, the production of vanillin from synthetic sources accounts for 95%, with 15% of the synthetic vanillin being obtained from LIG [114]. Several approaches have been devised to synthesize vanillin using Kraft LIG [115,116] and ferulic acid [117,118]. Trimethoprim is an antibiotic prescribed for urinary infections, while L-DOPA is used to manage Parkinson’s disease, as it serves as a precursor for the neurotransmitter dopamine [119]. Another study also investigated the potential of vanillin to protect against diabetic nephropathy, a prevalent consequence of diabetes that results in impaired kidney function [120]. In Figure 4, vanillin’s structure is represented.

A brief schematic of vanillin synthesis from LIG is represented in Figure 5.

Tablets represent the most prevalent pharmacological dose form [122]. They are very easy to produce, have commendable physical stability, and are widely accepted by patients [123]. A variety of medicinal excipients, including several polymers, can be utilized for direct compression [124,125,126]. This category of polymers encompasses synthetic macromolecules, like poly(vinyl pyrrolidone) and poly(acrylic acid), as well as natural polymers like cellulose [127]. Cellulose is a crucial excipient in tableting owing to its superior binding capabilities in the dry state [128]. Furthermore, cellulose is the most prevalent natural polymer on the planet. This biopolymer is found in plant cell walls and is therefore a renewable resource. In addition to cellulose and its derivatives, most excipients utilized in solid oral dose forms are synthetic polymers [129]. It is necessary to identify novel renewable polymers suitable for pharmaceutical applications [130]. Given that the pharmaceutical excipient industry is projected to reach USD 8.53 billion in value by 2023, significant endeavors have been undertaken to innovate new excipients for tablet formulation. LIG (LIG) is an exemplary renewable and economical option for this purpose. LIG is a biopolymer found in the cell walls of vascular plants, composed of randomly crosslinked networks of methoxylated and hydroxylated phenylpropane [131,132]. This chemical offers mechanical protection to the plant. Furthermore, LIG safeguards plants from external biological and chemical stressors due to its antioxidant and antibacterial characteristics [133,134]. Lignin is one of the most prevalent polymers on Earth, ranking second to cellulose. The primary distinction between cellulose and lignin is that the latter is comparatively underutilized [135,136]. Approximately 70 million tons of lignin produced during cellulose extraction by the paper industry are predominantly incinerated as low-grade fuel or dumped as trash [137,138]. Fewer than 2% of the total LIG generated is repurposed for the creation of niche items [139]. LIG possesses significant potential for application in novel functional and eco-friendly materials, owing to its abundance and beneficial qualities, including antioxidant and antibacterial activity. Over the past decade, researchers have exerted considerable effort to produce novel LIG-based materials [140,141]. This biopolymer has been utilized in numerous applications, including antibacterial agents, antioxidant additives, UV protection agents, hydrogel-forming molecules, nanoparticle components, and binders in lithium batteries, among others [142,143]. Nonetheless, the application of LIG as an excipient in pharmaceutical formulations is limited, with only a few studies documenting its utilization [144,145,146]. Consequently, additional research is required to augment the data presented in these studies and comprehensively grasp the potential of LIG as a pharmaceutical excipient. This study proposes the utilization of LIG as an excipient for direct compression in the formulation of drug-containing tablets. A model medicine, tetracycline (TC), was selected and mixed with LIG to formulate tablets. Furthermore, LIG was amalgamated with microcrystalline cellulose (MCC) to manufacture various tablet forms. The tablets were assessed based on their crushing strength, content uniformity, shape, wettability, antioxidant characteristics, and drug release profile [147,148].

There is a pressing necessity to discover renewable biopolymers that can replace synthetic ones. Natural biopolymers, such as cellulose and its derivatives, are widely utilized by the pharmaceutical sector. Currently, a diverse range of synthetic pharmaceutical excipients are used. Complete substitution of synthetic biopolymers with natural ones is unattainable because of specific constraints inherent in natural biopolymers, such as their intrinsic unpredictability. Nevertheless, LIG can serve as a pharmaceutical excipient for tablets, not limited to pharmaceutical applications. The scientific community is diligently striving to develop natural alternatives to address this deficiency. In addition, LIG can be used as an excipient in the preparation of dietary supplements or fertilizers in tablet form. Therefore, LIG has demonstrated intriguing characteristics, and as a result, we feel that its potential as a pharmaceutical excipient should be fully utilized for various applications [149,150].

### 4.10. Cosmetics

LIG’s antioxidant qualities make it highly important in skincare products. It effectively shields against oxidative stress induced by environmental elements, including UV radiation and pollution. LIG is a valuable component in anti-aging lotions, serums, and sunscreens. In addition, LIG’s emulsifying and thickening capabilities improve the texture and durability of cosmetic products (Figure 6).

### 4.11. Pharmaceutical Formulation

Administration of vanillin at a dose of 100 mg/kg, together with fasting blood glucose level, resulted in enhanced kidney function. The researchers determined that vanillin treatment demonstrated a strong protective effect on the kidneys against diabetic nephropathy. They recommend that administering vanillin at the initial stages of diabetic nephropathy should be a priority for future clinical studies involving humans. Several research studies in the literature have examined the anticancer properties of vanillin and compounds derived from vanillin. A prior study investigated the in vivo anticancer effects of vanillin semi-carbazone on Ehrlich ascites carcinoma cells in Swiss albino mice. Vanillic acid, also known as 4-hydroxy-3-methoxybenzoic acid, is a derivative of vanillin that has undergone oxidation. It is commonly employed as a flavoring ingredient. An animal model was utilized to explore the impact of vanillic acid, similar to vanillin, on the harmful effects of cisplatin, a widely used cancer medication [151,152]. This study demonstrated that male albino rats treated with vanillic acid at doses of 50–100 mg/kg exhibited a significant improvement in renal function and a reduction in antioxidant status, bringing them closer to normal levels. This effect was shown when comparing the vanillic acid-treated group to the group of animals treated alone with cisplatin. The results indicate that vanillin and vanillic acid have the potential to be utilized together as a combined treatment in cancer therapy [153].

Ferulic acid, also known as 4-hydroxy-3-methoxycinnamic acid, is a phenolic acid derived from LIG. It can be utilized in the production of vanillin and vanillic acid. Ferulic acid is commonly cross-linked with hemicelluloses through ester linkages in the plant cell wall [154]. The substance can be obtained through the use of hot water [155], deep eutectic solvents [156], or alkaline procedures [157]. It has been traditionally utilized in Chinese medicine to treat cardiovascular and cerebrovascular ailments [158]. Being a natural antioxidant, it can eliminate free radicals and possesses a diverse range of activities, including antioxidant, antibacterial, anti-inflammatory, antidiabetic, and anti-carcinogenic properties [159,160] (Figure 7).

Curcumin, a compound formed by the combination of two molecules of ferulic acid, specifically affects essential genes related to the growth of new blood vessels, programmed cell death, cell division, and the spread of cancer cells. Because of these effects, it is regarded as a substance that can fight against cancer [119,120,161]. Lin et al. conducted a study to examine the effects of ferulic acid on human keratinocyte HaCaT cells that were exposed to UVB radiation [122]. The analysis has shown that ferulic acid can suppress the formation of UVB-induced skin tumors and has potential anti-carcinogenic capabilities. A separate study examined the radiosensitizing effect of ferulic acid, which enhances the deadly effects of radiation, on two types of cervical cancer cells (HeLa and ME-180) [127]. A study revealed that ferulic acid intensifies the deleterious impact of radiation, leading to a reduction in cell viability and survival rate. Fahrioğlu et al. investigated the effect of ferulic acid on gene expression, cell survival, colony formation, and invasion in MIA PaCa-2 human pancreatic cancer cells [162]. According to their findings, ferulic acid acts as an anticancer agent by influencing the cell cycle, apoptosis, invasion, and colony formation of cancer cells.

In addition, only a limited number of studies have examined the impact of ferulic acid and its synergistic effects with other antioxidants on diabetes. Song et al. experimented to evaluate the efficacy of ferulic acid on rats that were both obese and diabetic [134]. It was discovered that it greatly enhanced the antioxidant activity in the plasma, heart, and liver. In addition, they documented the efficacy of their treatment in mitigating oxidative stress in obese rats suffering from advanced diabetes. A separate investigation examined the potential benefits of ferulic acid in mitigating protein glycation, lipid peroxidation, membrane ion pump activity, and phosphatidylserine exposure in human erythrocytes exposed to high glucose levels. The results demonstrate that ferulic acid can enhance the effects of hyperglycemia and avoid vascular damage linked to diabetes [163].

Coumaric acid is a compound that is derived from cinnamic acid and contains hydroxyl groups. In nature, the most common form of coumaric acid is known as ρ-coumaric acid. Alkaline hydrolysis is a technique that may be employed to produce it, and it can mitigate the detrimental impact of UV radiation on cells. This is why it is frequently utilized as a critical component in cosmetics. Coumaric acid, similarly to ferulic acid, is a well-known plant-derived antioxidant. The antioxidant activity of the substance was evaluated alongside other phenolic compounds, including ferulic acid and caffeic acid, on various occasions. Yeh et al. conducted a study on the lipid-lowering and antioxidative effects of ρ-coumaric acid, ferulic acid, and caffeic acid [164]. In a recent study, the antidiabetic benefits of 11 phenolic acids, including ρ-coumaric acid, were compared to metformin [120]. The findings demonstrated that all phenolic acids showed similar or more potent effects on glucose absorption in HepG2 cells [165].

Additionally, this study discovered that coumaric acid exhibits one of the most potent inhibitory effects on glucosidase among the three phenolic acids. The study examined the potential preventive properties of ρ-coumaric acid and ferulic acid against colon cancer utilizing the Caco-2 endothelial tumor cell line. The study discovered that both of these chemicals demonstrated anti-proliferative actions on Caco-2 human cancer cells and decreased the number of cancer cells to 43–75% of the control after 2–3 days of treatment. Roy et al. investigated the impact of ρ-coumaric acid and ferulic acid on the HCT 15 human colorectal cancer cell line and the epidermal growth factor receptor (EGFR), which is believed to have a substantial influence on the progression of colorectal cancer. It was found that several chemicals can impede the action of EGFR at its active site. Additionally, the cytotoxicity experiments revealed that both ρ-coumaric acid and ferulic acid exhibited significant effectiveness in triggering cell death in colorectal cancer cells [166].

Syringic acid (4-hydroxy-3,5-dimethoxybenzoic acid) is a phenolic compound recognized for its potent antioxidant properties and may be synthesized through alkaline hydrolysis [167]. It may serve as a therapeutic agent for numerous conditions, including diabetes, cancer, and hepatic damage [168]. It can regulate the dynamics of various biological targets, including transcriptional and growth factors [169]. The foliage and bark of various Quercus species (a little oak tree) have been utilized to extract syringic acid and other phenolic compounds for the evaluation of their biological activities [170]. Quercus infectoria is a renowned traditional medicine in Asia, utilized for the treatment of wound infections and toothache. In 1979, syringic acid, derived from powdered galls of Quercus infectoria using solvent extraction, and the neuropharmacological effects of a syringic acid-rich extract were evaluated in mice [171]. The antibacterial efficacy of syringic acid and plant extracts containing syringic acid was evaluated against various bacteria and fungi [172,173]. Abaza et al. examined the antimitogenic and chemosensitizing properties of syringic acid derived from Tamarix aucheriana (salt cedar plant) against human colorectal cancer cell lines SW1116 and SW837 [174]. Syringic acid had a time- and dose-dependent antimitogenic impact on cancer cells, demonstrating no damage towards normal fibroblasts. It was also claimed that it enhanced the susceptibility of cancer cells to conventional chemotherapies, increasing their responsiveness by up to 20,000-fold relative to regular medications.

Eugenol, also known as 4-allyl-2-methoxyphenol, is a chemical compound that is obtained from LIG found in woody biomass. Eugenol can undergo many metabolic routes to be transformed into ferulic acid and vanillin [175,176,177]. LIG depolymerization can yield not just eugenol itself but also a diverse range of active biomaterials. According to Varanasi et al., the production of chemicals derived from LIG, such as eugenol, phenols, guaiacols, syringes, and catechols, relies on the initial concentration of biomass and the temperature at which it is dissolved. They were able to produce approximately 2 g of eugenol from 1 kg of low sulfonate alkali LIG by dissolving it at 160 °C for 6 h with a biomass loading of 3% [178]. Eugenol is commonly employed as a flavoring ingredient and food additive (Figure 8).

Additionally, the literature contains several studies that investigate the antioxidant and antibacterial properties of the subject. Experiments have been conducted to assess the efficacy of eugenol and isoeugenol against various foodborne pathogens, including *S. aureus*, *Bacillus subtilis*, *Listeria monocytogenes*, *E. coli*, *Salmonella typhimurium*, and *Shigella dysenteriae* [179]. A study demonstrated that exposure to eugenol rescued SHSY5Y cells from glucose-induced cell death and enhanced cell survival. The animal-based model demonstrated that eugenol administration had a significant impact on reducing the average body weight and blood glucose levels of diabetic rats. A separate study conducted on animals showed that eugenol, a compound found in animals, has anti-diabetic solid properties. This was evidenced by a notable decrease in serum glucose, triglyceride, and cholesterol levels in diabetic male adult Sprague Dawley rats [180,181,182]. Furthermore, this study showed that administering eugenol at a dose of 10–20 mg/kg improved insulin sensitivity. This finding suggests that eugenol has potential as a therapeutic agent for the prevention of type 2 diabetes.

Lignophenol, a functional polymer derived from LIG, can be obtained by performing phase separation processes using phenol derivatives and concentrated acid [183]. Despite the fact that lignophenols possess significant antioxidant characteristics, their physiological function and possible medicinal applications have not been thoroughly described [184,185,186]. The literature has documented the medicinal potential of lignophenols through in vitro and animal-based investigations. According to a study conducted on rats with streptozotocin-induced diabetes, lignophenols were found to reduce oxidative and inflammatory harm in the kidney. This was achieved by inhibiting excessive oxidative stress and the inflammation and activation of macrophages in the diabetic kidney [187]. Lignophenols were found to be crucial in enhancing vascular function in individuals with diabetes by reducing oxidative stress and inflammation in blood vessels through the inhibition of NAD(P)H oxidase, as demonstrated in a separate research investigation [188,189,190]. These findings suggest that lignophenols can regulate the prevalent diseases of the modern era, namely diabetes and obesity (Figure 9).

The primary by-products of the sulfite pulping process are water-soluble lignosulfonates, which are salts derived from lignosulfonic acid. These materials have been demonstrated to be valuable raw resources for fine compounds, such as vanillin [191]. Lignosulfonic acid (LSA) is a polyanionic macromolecule derived from LIG, a low-cost by-product of the pulp and paper industries. The acknowledged antiviral and antibacterial efficacy of LSA underscores its potential as an economical medicinal agent. Gordts et al. conducted experiments to examine the antiviral effects of pure LSA (a commercially available substance) against HIV and HSV in several cellular tests [192]. The researchers showed that LSA effectively prevented the infection of T cells by HIV and HSV. Additionally, LSA had potent inhibitory effects on the replication of HIV. Additionally, they stated that LSA specifically targeted the proteins on the outer layer of the virus and did not exhibit any antiviral effects on viruses that lack an outer layer. Several studies in the literature have explored the potential of LSA for controlled drug release. Microspheres composed of a combination of LSA and gelatine were created by cross-linking with glutaraldehyde. These microspheres were utilized to achieve controlled release of an anti-malarial medication [193]. This study showed that the presence of microspheres increased the pace at which the medication was released for a duration of up to 10 h. Furthermore, the release of the drug was found to be influenced by the pH levels. A study was conducted using LSA and sodium alginate mix microspheres to create a polymer matrix that allows for the controlled release of an antibiotic called ciprofloxacin [194]. According to the findings, the carrier that was created is appropriate for delivering drugs in a controlled manner for gastrointestinal purposes. Hasegawa et al. investigated the inhibitory effects of LSA on the absorption of glucose in the intestines [195]. In human colorectal cancer cells, it was shown that LSA hindered the uptake of 2-deoxyglucose. In their rat experiments, it was observed that LSA effectively regulated the increase in blood glucose levels. Feeding diabetic KK-Ay mice with LSA significantly reduced the growth in serum glucose levels by inhibiting α-glucose activity and intestinal glucose absorption [196]. These findings indicate that in addition to lignophenols, LSA may have the potential to manage obesity and diabetes. Complexes formed between LIG and carbohydrates, LIG–carbohydrate complexes (LCCs), are formed in the cell walls of lignified plants through covalent linkages between certain polysaccharides and LIG [197]. There are a total of eight distinct types of links between LIG and carbohydrates. These include benzyl ether, benzyl ester, glycosidic, phenyl glycosidic, hemiacetal linkages, acetal linkages, ferulate ester, and ferulate ester bonds [198]. The primary forms of LCC links in wood include phenyl glycoside, benzyl ethers, and ester linkages [199]. In contrast, non-woody plants mostly contain ferulate and ferulate esters as prominent LCC connections [199]. Under acidic conditions, the benzyl ethers and phenyl glycoside bonds present in wood can be readily broken [140]. The variable content and structure of natural lignocellulosic composites might be attributed to the presence of various forms of LIG and polysaccharides in different lignocellulosic biomasses. The existence of LCC, whether it occurs naturally or is produced during processing, is regarded as a contributing factor to the challenges encountered in the chemical and biological treatment of lignocellulosic biomass. A study conducted by researchers recovered six different types of LCC fractions from Eucalyptus using a combination of aqueous dioxane and successive precipitation with 70% ethanol, 100% ethanol, and acidic water. The study demonstrated that the low molecular weight LCC, which had a significant carbohydrate content (60–63%), was separated during the initial extraction process using 70% ethanol. The primary structures identified in the LCCs recovered from poplar’s hot water pretreatment liquid were glucomannan-LIG and glucuronoxylan-LIG, as shown in a recently published work [200]. Biomass conversion and biorefining have shown the ability to stimulate antibiotic activity in mice afflicted with Staphylococcus aureus, a pathogenic bacterium responsible for a diverse range of clinical illnesses [201]. Another study found that unrefined LIG, which was obtained by using an alkaline extraction process on leftover maize stover from ethanol production, showed antibacterial properties against the Gram-positive bacteria *S. aureus* and *Listeria monocytogenes* [202].

However, the extracts did not exhibit the same effect on Gram-negative bacteria, such as *E. coli* and *S. enteritidis*. The study also found that the antibacterial properties of the extracts were in line with their antioxidant properties, which were similarly influenced by the extraction conditions, such as temperature and the ratio of residue to solvent. Low-cost carriers have also been utilized for anticancer research. Sakagami et al. conducted a study to examine the anticancer properties of LCCs derived from hot water and alkaline extracts of pine cones, based on traditional knowledge suggesting their effectiveness against gastroenterological cancers. Researchers discovered evidence indicating that isolated LCCs greatly extended the lifespan of mice that had received implants of ascites tumor cells (sarcoma-180). Inonotus obliquus, also known as Chaga mushroom, is a traditional medicine that has been utilized for the treatment of various malignant tumors in humans since the sixteenth century [203,204]. Niu et al. conducted a study to examine the properties, as well as the antioxidant and immunological activities, in a laboratory setting, of LCCs that were obtained from the alkaline extract of I. obliquus [205]. They stated that extracts with various radical scavenging capabilities showed outstanding antioxidant and immunological properties. These findings indicate that certain LCCs may be responsible for the renowned anti-tumor effects observed in certain plants. LCCs are employed as a natural UV-blocking ingredient in sunscreens and moisturizers. The UV protection efficacy of resveratrol and vitamin C was compared with the LCCs derived from Lentinus edodes mycelia. The findings indicated that the anti-UV effectiveness of LCCs was similar to that of two widely recognized UV-protective chemicals [206]. A separate investigation demonstrated that low molecular weight compounds (LCCs) obtained from pine cone and pine seed shells exhibited remarkable effectiveness in protecting against ultraviolet (UV) radiation. These LCCs were isolated by a series of alkaline extraction and acid precipitation procedures [207].

Hydrogels are typically described as hydrophilic polymers that form a three-dimensional structure capable of holding a significant amount of water. Advantageous features often encompass characteristics such as non-toxicity, high capacity for drug loading, ability to degrade naturally, compatibility with living organisms, exceptional support structure, and a well-organized arrangement [198]. Hydrogels possessing these characteristics hold promise for use in personal hygiene items, medication delivery devices, wound healing dressings, and regenerative medical treatments [134,208,209]. The utilization of natural polymers for hydrogel creation has experienced a growing trend in recent years [210]. Hyaluronic acid, chondroitin sulfate, chitosan, gelatine, alginate, and cellulose derivatives have been utilized in the creation of hydrogel systems based on biopolymers [211,212]. LIG possesses considerable promise for use in the production of biodegradable hydrogels. It is rich in functional hydrophilic groups such as hydroxyls, carbonyls, and methoxyls, which enable straightforward chemical modification for various purposes [201]. LIG possesses several inherent benefits, including antibacterial, antioxidant, and biodegradable characteristics. Therefore, hydrogels derived from LIG exhibit favorable characteristics as coverings for medicinal materials [213].

The three primary techniques employed for the synthesis of LIG-based hydrogels are cross-linking copolymerization, cross-linking of reactive polymer precursors, and cross-linking via polymer–polymer interaction. Elsewhere, the various synthetic methods and cross-linkers employed in the development of hydrogels based on LIG have been thoroughly examined [213]. The researchers created biocompatible hydrogels by combining a 2.5% (*w*/*v*) chitosan solution in acetic acid with a 10% (*w*/*v*) alkali LIG solution. The resulting gels were found to be non-toxic to both stem cells and animals. Based on these findings, the authors concluded that the cross-linked products have significant potential for use in wound healing applications [214]. Mondal et al. have created a hydrogel with exceptional antibacterial properties and rapid self-healing capabilities by utilizing a significant quantity of lignosulfonate and Al+3. A separate investigation was conducted to assess the mechanical durability and compatibility with living organisms of hydrogels made from hyaluronan and Kraft LIG, which were bonded together using carbodiimide. According to the scientists, the inclusion of Kraft LIG in amounts of up to 3% (*w*/*w*) enhanced the durability of the hydrogels [215]. Raschip et al. created hydrogel films by combining LIG derived from annual fiber crops with xanthan gums, which are commonly used as a food additive and thickening agent. The purpose of this was to release vanillin. The researchers discovered that the LIG served as an antioxidant agent and enhanced the biocompatibility of the resulting hydrogels [216]. Recently, a soluble fraction of LIG was extracted and separated from the black liquor oil of empty fruit bunches using an acidification process. This isolated LIG was then utilized in the synthesis of a LIG-agarose hydrogel, with epichlorohydrin serving as the cross-linking agent [217]. The study indicated that the hydrogels that were created possess favorable mechanical characteristics. A separate investigation involved the creation of hydrogels by means of the radical polymerization of hardwood Kraft LIG, which was then compared to synthetic hydrogels. The study demonstrated that hydrogels derived from LIG exhibit a greater capacity for swelling and superior thermal stability compared to synthetic hydrogels. The production of hydrogels based on LIG is still an emerging research field, and there are just a few therapeutic trials available.

### 4.12. Role in Sustainable Development: Circular Economy

LIG is often considered a waste product in the paper and pulp industry. However, its valorization into high-value products such as chemicals, biofuels, and materials aligns with the principles of the circular economy, where waste is minimized and resources are maximized. By transforming LIG into valuable products, industries can reduce their environmental impact and contribute to more sustainable production processes. LIG is central to the development of a bio-based economy, where renewable biological resources replace fossil-based materials. Its utilization in creating biofuels, biochemicals, and biomaterials supports the reduction of carbon footprints and promotes sustainability. The ongoing research in LIG valorization is pivotal in advancing circular economy practices, where waste materials are transformed into valuable products [218].

Podophyllotoxin, a lignan derived from podophyllum species, has been shown to possess various types of pharmaceutical activity, such as anthelmintic, antifungal, antiviral, and antineoplastic properties. Previous reports have demonstrated that PPT and its derivatives, including etoposide and teniposide, have been successfully utilized to treat lung cancer, liver cancer, breast cancer, non-Hodgkin and other lymphomas. The mechanism of the anti-cancer activities of PPT is mainly attributed to the binding of the colchicine site of tubulin, disrupting microtubule assembly, which results in mitotic arrest and cellular apoptosis. However, the systemic application of PPT for the treatment of cancer has been greatly limited due to poor water solubility and lack of selectivity. Therefore, it is of critical importance to develop a treatment strategy that can improve the aqueous solubility and selectivity of PPT. The use of delivery systems to improve the water solubility of lipophilic drugs has been explored during the past several decades. Among those delivery systems, cyclodextrin (CD) complexation has become the focus of interest for hydrophobic drug delivery due to its reliable safety profile, simple preparation method, and high drug loading capacity. Cyclodextrins (CDs) are cyclic derivatives of starch that are obtained from starch by enzymatic process. They are torus-shaped circular α-(1,4) linked oligosaccharides that have been extensively used to improve the aqueous solubility, bioavailability, and stability or decrease unfavorable side effects of drugs. The glucose chains in CDs form a unique conical structure with a hydrophobic cavity, and lipophilic compounds may enter and form water-soluble complexes that alter the physical and chemical properties of the drug. α-, β- and γ-CDs consist of six, seven, and eight glucose units, respectively, and are the most studied cyclodextrins. In particular, β-CD is more extensively used in drug delivery systems due to its appropriate cavity size, good ability to combine aromatic units, ready availability, easy production, and relatively economical price. However, the low water solubility (1.85 g/100 mL) of parent β-CD limits its further application in pharmaceutical formulations. The relatively low water solubility of β-CD may be due to an internal hydrogen bond formed between the C-2-OH and the C-3-OH of the neighboring glucose unit. The formation of the hydrogen bond in the β-CD molecule results in a secondary belt, leading to a relatively rigid structure [219]. In addition, β-CD application is also limited due to the lack of selectivity. The development of a site-specific delivery system with greater efficacy and lower toxicity has recently become an urgent need to overcome the limitations of conventional therapy [220] (Figure 10).

Biotin, one of the B vitamins, also known as vitamin H, is a water-soluble vitamin. As a cellular growth promoter, biotin and its derivatives have already been used in the field of cancer studies and tissue engineering. Biotin was found in the kidney, liver, pancreas, and milk. Due to the rapid cell growth and enhanced proliferation, cancer cells need more of certain vitamins than normal cells. Therefore, the receptors involved in the uptake of vitamins are usually overexpressed on the surface of tumor cells. As a consequence, these surface receptors are helpful as tumor-targeting biomarkers. It has been reported that additional biotin is needed for the rapid growth and proliferation of cancer cells. Specifically, biotin is present in higher content in cancerous tissue than in normal tissue. Coincidentally, biotin receptors have been reported to be over-expressed on the surfaces of many types of tumor cells. Highly proliferating cancer cells, such as MDA-M231, MCF7, A549, HeLa, and HepG2 cells, exhibit elevated biotin receptors in comparison with healthy cells. Therefore, biotin is a famous targeting agent for drug delivery systems. As a specific active targeting agent, biotin has been utilized in drug carriers to increase intra-cellular uptake of drugs and decrease toxicity in normal tissues. When biotin is conjugated with other drugs via amide or ester linkages, it spontaneously acts as a targeting moiety for specific interaction with tumor cells. Previous reports demonstrated that a biotin and arginine-modified hydroxypropyl-β-cyclodextrin could improve the anticancer activity of paclitaxel [221].

Therefore, we hypothesized that biotin as a tumor-specific ligand conjugated with β-CD to strengthen its cancer selectivity is feasible. The purpose of this study is to improve the water solubility and cancer selectivity of the PPT through the formation of PPT/B-CD inclusion complexes. The inclusion complexes of PPT/B-CD were prepared and analyzed by water solubility, phase solubility, Job’s plot, 1 H NMR and 2D ROESY NMR, powder X-ray diffraction (XRD), Fourier transform infrared spectroscopy (FT-IR), and scanning electron microscopy (SEM). In addition, the cell cytotoxicity experiment was conducted to study the antitumor activity of the PPT/B-CD complexes. The cellular uptake was carried out to investigate the targeting ability of B-CD with rhodamine B as a fluorescence probe [222].

### 4.13. Renewable Raw Material

As a naturally occurring polymer, LIG’s application as a renewable raw material is critical in the transition to a bio-based economy. Its use in replacing fossil-derived materials helps reduce carbon footprints and promotes sustainable development. Industries are increasingly focusing on LIG as a source for producing bio-based chemicals, materials, and energy, supporting global efforts to mitigate climate change. LIG’s role in carbon sequestration is also significant. By contributing to the long-term storage of carbon in soil and plant biomass, LIG helps mitigate climate change. Forests, where LIG-rich biomass is abundant, act as carbon sinks, absorbing more carbon dioxide than they release, which is crucial in the global effort to reduce atmospheric carbon levels [223].

LIG, which is a non-edible part of biomass, contains valuable functional groups that are sought after for chemical syntheses. Efficiently breaking down LIG while preserving the precious cellulose and hemicellulose has been a major obstacle. Current biomass processing methods either result in significant condensation of LIG, which makes it challenging to use chemically, or prioritize complete depolymerization of LIG to generate monomers that are hard to separate for subsequent chemical synthesis. In this study, we present a novel method for selectively breaking down polymers, resulting in the formation of oligomers that can be easily transformed into polymer networks that are chemically recyclable. The technique exploits the high specificity of photocatalytic activation of the β-O-4 bond in LIG using tetrabutylammonium decatungstate (TBADT). The presence of external electron mediators or scavengers facilitates the breaking or oxidation of this bond, respectively, allowing for precise control over the depolymerization process and the concentration of a crucial functional group, C-O, in the resulting products [224].

As an important component of lignocellulose, LIG offers numerous advantages as an attractive feedstock. For example, LIG is abundant, accounting for 15−40% of the total biomass; it is rich in aromatic functionalities that are of great potential value for chemical synthesis and material fabrication; LIG is inedible, so its utilization will not compete with food needs. However, existing biomass processing technologies prioritize cellulose and hemicellulose. As a result, LIG has been significantly underutilized. Consider the traditional pulping process as an example. The delignification methods produce the so-called technical LIG, which often leads to structural heterogeneity and undesired side reactions (e.g., condensation) and makes its subsequent chemical utilization challenging. Recently, an alternative LIG-first strategy has emerged to convert native LIG in lignocellulose into value-added chemicals directly. For instance, reductive catalytic fractionation (RCF), as a LIG-first approach, produces a mixture of low molecular weight compounds from native LIG. However, the mixture produced by RCF is often complex to separate. Moreover, RCF tends to destroy high-value functional groups such as carboxylic acids, aldehydes, and aromatic rings, undermining the value of these products as precursors for chemical syntheses. Indeed, most RCF studies focus on retrieving the thermal energy of the products by using them as fuels. Recognizing these challenges, researchers have recently turned their attention to depolymerizing native LIG under mild conditions. Successful examples have been demonstrated to utilize the hydrogen-atom transfer (HAT) reaction for selectively targeting the abundant β-1 and β-O-4 motifs. A unique advantage offered by HAT is the ability to preserve the aromatics, ketones, and aldehydes. Nevertheless, earlier attempts at using HAT-based chemistries for LIG valorization have primarily focused on producing small molecules, which remain challenging to separate. On the other hand, partial depolymerization of LIG has started to show its promise for the construction of functional materials, such as thermoset plastics, elastomers, or trimers. However, these initial materials are constructed from kraft LIG, which has already undergone significant unwanted chemical modifications in the pulping process that affect its chemical integrity [225].

## 5. Conclusions

LIG, an abundant and sustainable resource, contains a plethora of aromatic chemicals. LIG and LIG-derived aromatics can be used as starting materials to synthesize aromatic compounds, ring-cleaved products, and bioactive molecules with added value. The findings suggest that synthetic biology can facilitate the conversion of LIG into valuable compounds. The primary problem in LIG valorization lies in the structural diversity and variability of the available technological LIGs. LIGs are regarded as significant compounds in anti-tumor therapies; several studies have also examined polyphenols with structural characteristics present in LIG mixes. The cytotoxic activity of LIG and its derivatives is frequently attributed to interactions between LIG and other substances or LIG-containing natural complexes, such as LIG–LIG-carbohydrate complexes. The influence of LIG modification on drug release and pH-dependent releasing behavior of oral solid dosage forms is used now in different formulations. LIG can be readily chemically modified due to the existence of various functional groups in its structure. That is why using biodegradable polymers in the pharmaceutical field is crucial.

Notwithstanding the structural obstacles related to regulatory considerations, the suitability of LIG as an excipient for conventional tablet manufacture has been evaluated and documented. The integration of medicinal compounds with LIG or chemically synthesized carboxylated LIG constitutes novel pharmaceutical dosage formulations. With respect to applications in the pharmaceutical sector, the usage of LIG particles as delivery vehicles is most apparent; such actives would have to be co-precipitated upon particle formation, requiring a comparable solubility profile. This is a critical parameter for the formulation of nanoparticles. In fact, the presence of LIG increased the release efficiency while protecting the active substances. In terms of using LIG as the main active molecule, besides challenges similar to those just presented, more work in terms of structure–activity relationships and the release of active substances is needed and might require an innovative formulation.

## Figures and Tables

**Figure 1 pharmaceuticals-17-01406-f001:**
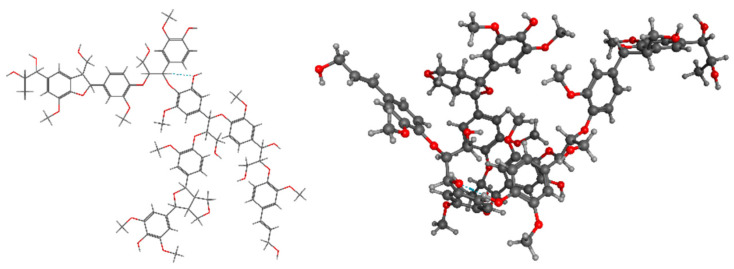
LIG structure is represented as balls and sticks.

**Figure 2 pharmaceuticals-17-01406-f002:**
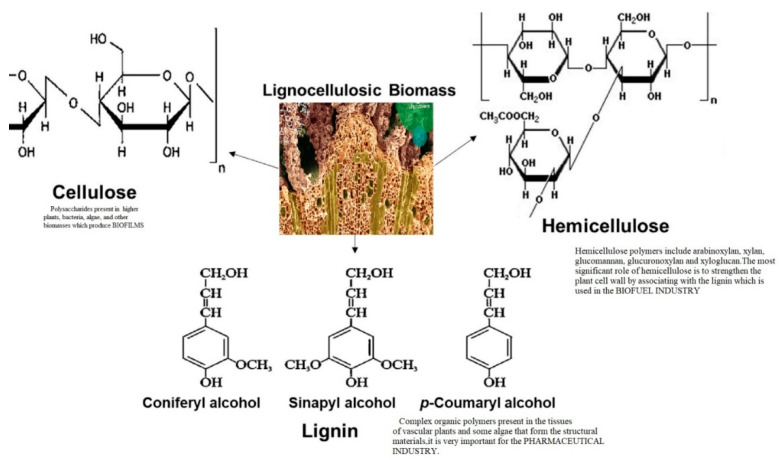
The structure of lignocellulosic biomass and its importance (picture—personal collection).

**Figure 3 pharmaceuticals-17-01406-f003:**
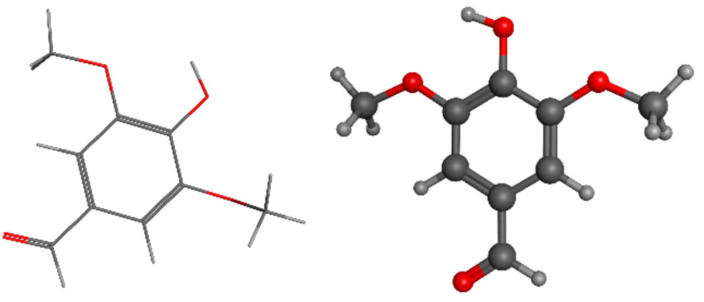
Syringaldehyde structure is represented as balls and sticks.

**Figure 4 pharmaceuticals-17-01406-f004:**
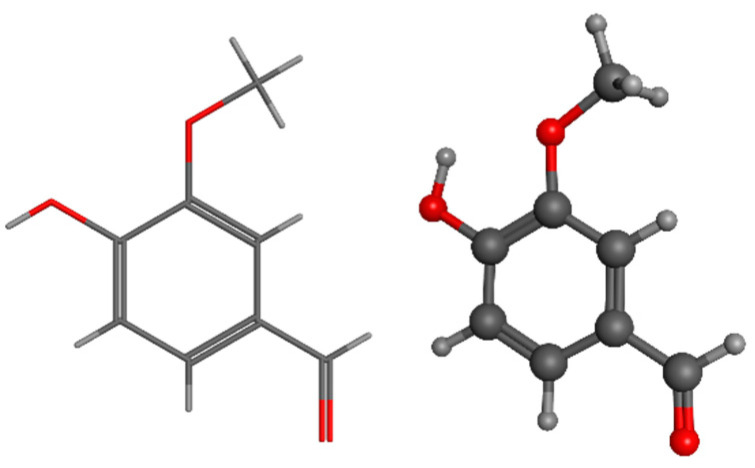
Vanillin structure is represented as balls and sticks.

**Figure 5 pharmaceuticals-17-01406-f005:**
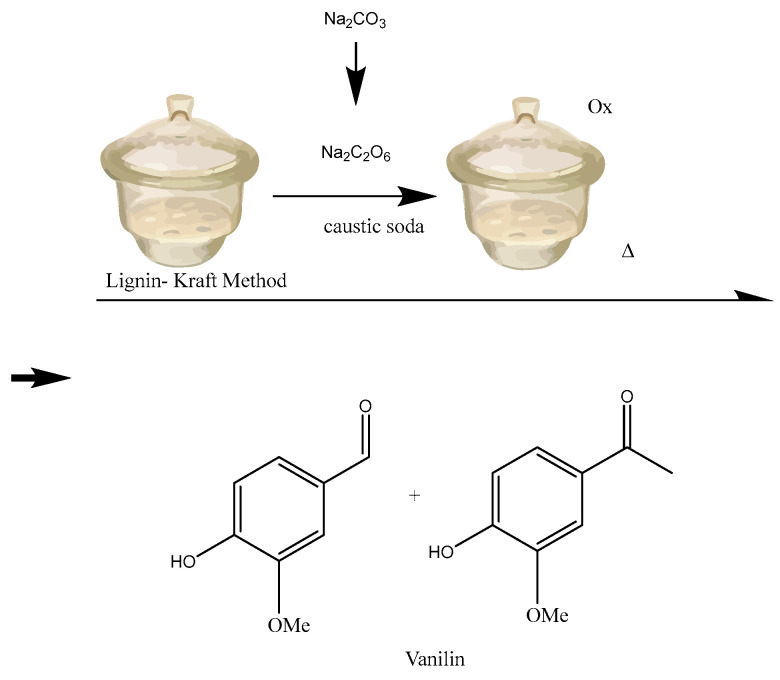
Vanillin synthesis from LIG. Lignin resulting from the Kraft method is exposed to caustic soda using Na2C2O6 as a “green” oxidizer [121].

**Figure 6 pharmaceuticals-17-01406-f006:**
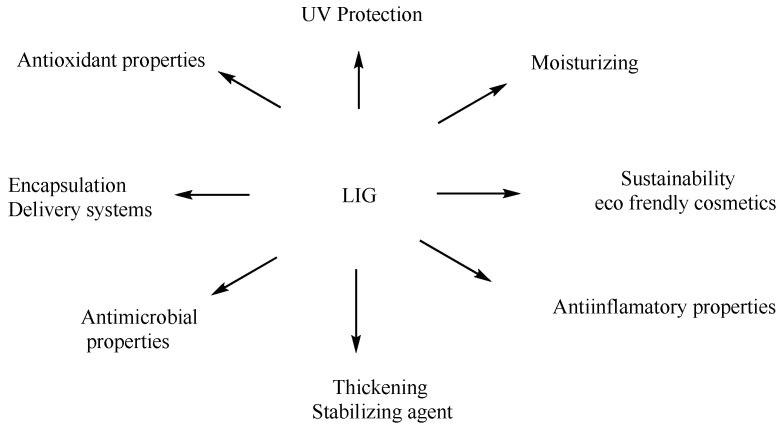
LIG used in cosmetics.

**Figure 7 pharmaceuticals-17-01406-f007:**
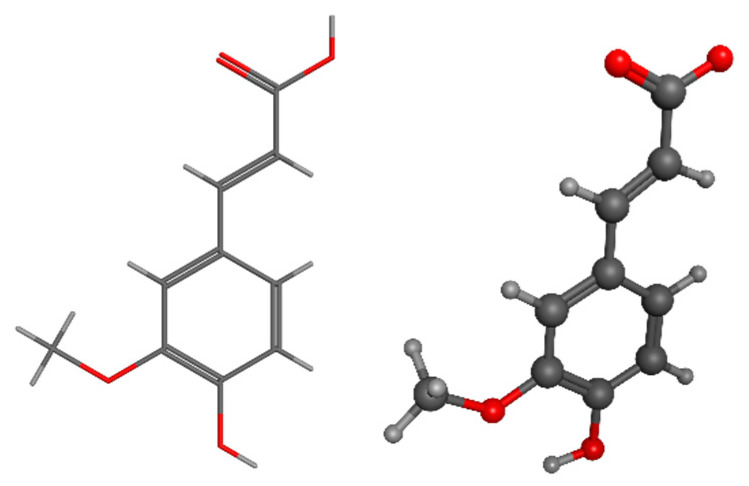
Ferulic acid structure is represented as balls and sticks.

**Figure 8 pharmaceuticals-17-01406-f008:**
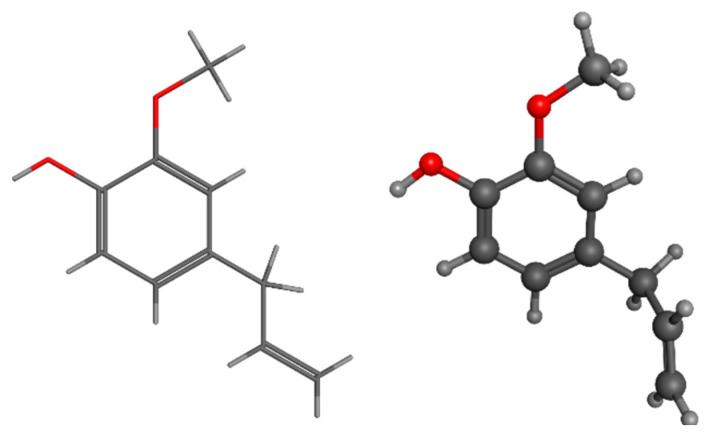
Eugenol structure is represented as balls and sticks.

**Figure 9 pharmaceuticals-17-01406-f009:**
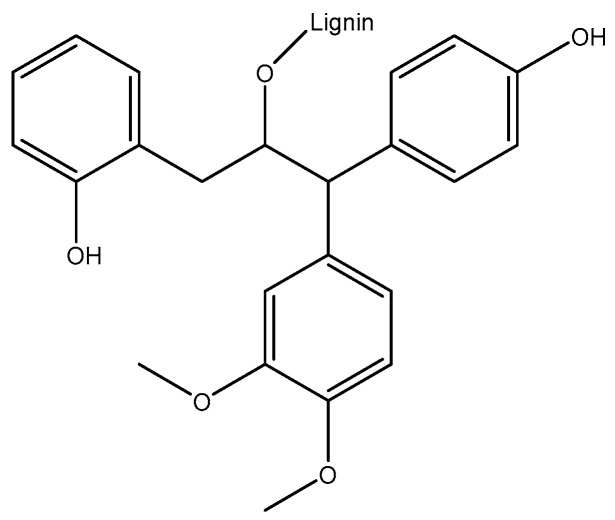
Lignophenol structure is represented.

**Figure 10 pharmaceuticals-17-01406-f010:**
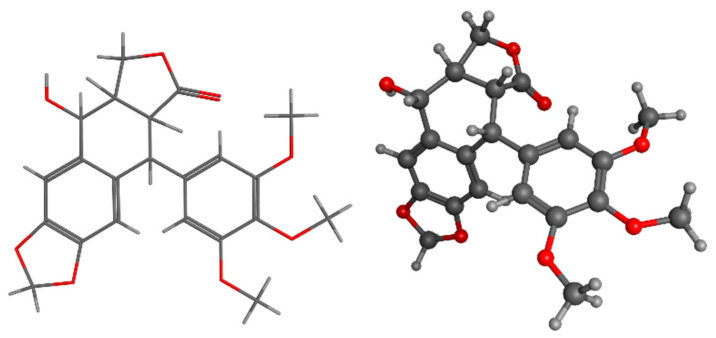
Podophyllotoxin, a lignan derived from the podophyllum species structure, is represented as balls and sticks.

**Table 1 pharmaceuticals-17-01406-t001:** LIG extraction methods.

Nr	Method	Description	Advantage	Ref.
1	Kraft Process	The Kraft process is the most widely used chemical pulping method for lignin extraction, especially in the paper industry. It uses a mixture of sodium hydroxide (NaOH) and sodium sulfide (Na2S) to break down lignin and separate it from cellulose fibers.	This method provides high lignin removal efficiency, but the resulting lignin is chemically modified and contains sulfur, limiting its use in high-value applications.	[16]
2	Organosolv Process	The Organosolv process involves the use of organic solvents (e.g., ethanol, acetone, or formic acid) to solubilize lignin, separating it from cellulose and hemicellulose. This method is often used to produce sulfur-free lignin, which has a broader range of applications compared to Kraft lignin.	Produces high-purity lignin with minimal chemical alteration, making it suitable for bio-based materials and chemicals.	[17]
3	Soda Process	The Soda process uses sodium hydroxide (NaOH) to extract lignin from biomass. This method is used mainly for non-wood materials such as agricultural residues.	This is a sulfur-free process, producing lignin that can be more efficiently utilized in applications like biopolymers and bio-composites.	[18]
4	Ionic Liquid-Based Extraction	Ionic liquids (ILs) are used as solvents to dissolve lignin selectively, offering an eco-friendly alternative to traditional solvents. ILs can be recovered and reused, making the process more sustainable.	This method is more environmentally friendly and has the potential to produce high-quality lignin with minimal degradation.	[19]
5	Enzymatic Lignin Extraction	Enzymatic methods use lignin-degrading enzymes, such as laccases or peroxidases, to selectively break down lignin, preserving the polysaccharide components of the biomass.	This method is environmentally friendly and produces lignin with minimal chemical modification, making it suitable for high-value applications.	[20]
6	Steam Explosion	In steam explosion, biomass is treated with high-pressure steam, which breaks down the lignin structure and makes it easier to separate from cellulose and hemicellulose.	It is a physical method that avoids the use of chemicals, reducing environmental impact. However, it may result in lignin with altered chemical properties.	[21]
7	Ammonia Fiber Expansion (AFEX)	AFEX involves treating lignocellulosic biomass with liquid ammonia under high pressure. This process effectively separates lignin from cellulose and hemicellulose.	It produces high-quality lignin suitable for bio-based materials while preserving the cellulose and hemicellulose components for further use.	[22]
8	Sulfur-Free Alkali Extraction	Sulfur-free alkali extraction is similar to the Kraft process but avoids the use of sulfur compounds. It uses sodium hydroxide (NaOH) or potassium hydroxide (KOH) to dissolve lignin. This method is often employed to produce lignin with fewer contaminants, improving its potential for bio-based applications.	Avoids sulfur contamination, making the resulting lignin more suitable for high-value applications such as carbon fiber and bioplastics.	[23]
9	Acid Hydrolysis	In acid hydrolysis, lignin is separated from cellulose and hemicellulose using concentrated acids (e.g., sulfuric acid or hydrochloric acid). This method is widely used in the production of cellulosic ethanol, where lignin is a byproduct.	Highly effective in breaking down lignin but can result in some chemical degradation. Lignin from acid hydrolysis is often used for energy generation or in low-value applications due to its altered properties.	[24]
10	Alkaline Peroxide Extraction	Alkaline peroxide (a mixture of hydrogen peroxide and sodium hydroxide) is used to break down lignin. The peroxide acts as an oxidizing agent, further depolymerizing lignin and making it easier to extract.	This results in relatively mild depolymerization of lignin, which produces a product that retains some of its native chemical structure.	[25]
11	Deep Eutectic Solvent (DES) Extraction	Deep eutectic solvents are a type of green solvent that can be used for lignin extraction. These solvents are formed by mixing a hydrogen bond donor and acceptor, which dissolve lignin while preserving cellulose and hemicellulose.	DESs are biodegradable, non-toxic, and cost-effective, making them an attractive option for sustainable lignin extraction.	[26]
12	Supercritical Fluid Extraction (SFE)	Supercritical fluids, particularly carbon dioxide (CO_2_), are used to extract lignin from biomass. In supercritical conditions, CO_2_ can act as both a gas and a liquid, efficiently solubilizing and separating lignin from other components.	This method avoids the use of harmful solvents and allows for efficient recovery of both lignin and hemicellulose, making it a green alternative for lignin extraction.	[27]
13	Ammonia Recycle Percolation (ARP)	The ARP process uses aqueous ammonia in a flow-through reactor to extract lignin from biomass. Ammonia helps dissolve lignin and hemicellulose while leaving cellulose relatively intact.	It produces high-purity lignin that is suitable for advanced materials and bio-based chemicals. The ammonia used in the process can be recovered and recycled, making the process more sustainable.	[28]
14	Oxidative Extraction	Oxidative methods for lignin extraction use oxidizing agents (e.g., oxygen, ozone, or hydrogen peroxide) to depolymerize lignin and separate it from cellulose. This process is typically carried out in alkaline conditions.	Results in highly functionalized lignin, which can be used as a precursor for bio-based chemicals, but the degree of oxidation must be controlled to avoid over-degradation.	[29]
15	Hydrothermal Extraction	Hydrothermal extraction uses water at high temperatures and pressures to break down the lignin structure, often in combination with mild acids or alkalis to enhance solubility.	This is a relatively simple and environmentally friendly process, producing lignin with a more native structure. It is often used in conjunction with other extraction methods to improve overall yield.	[30]
16	Dilute Acid Pretreatment	In this method, dilute acid (usually sulfuric or hydrochloric acid) is applied to biomass at elevated temperatures to break down lignin. It is commonly used in conjunction with enzymatic hydrolysis for bioethanol production.	It is effective at removing hemicellulose and loosening the lignin structure for subsequent separation, but the process can result in significant lignin modification.	[31]

**Table 2 pharmaceuticals-17-01406-t002:** LIG antimicrobial and antifungal mode of action (MOA).

#	Mechanism	Description	Ref.
1	Disruption of Cell Walls and Membranes	Cell Wall Integrity: LIG and its derivatives can interact with the components of microbial cell walls, particularly in bacteria. This interaction can compromise the integrity of the cell wall, leading to cell lysis and death. Membrane Permeability: LIG can embed itself into the lipid bilayer of microbial cell membranes, causing increased permeability. This disruption allows the leakage of essential intracellular contents and ultimately results in cell death.	[34]
2	Generation of Reactive Oxygen Species (ROS)	Oxidative Stress: LIG can induce the production of reactive oxygen species (ROS) within microbial cells. ROS are highly reactive molecules that can damage cellular components such as DNA, proteins, and lipids, leading to oxidative stress and cell death. The phenolic groups in LIG can undergo redox cycling, contributing to ROS generation. This mechanism is particularly effective against a broad range of microorganisms.	[35]
3	Enzyme Inhibition	Inhibition of Metabolic Enzymes: LIG can inhibit key enzymes involved in microbial metabolism. The phenolic compounds in LIG can bind to the active sites of these enzymes, preventing them from catalyzing essential biochemical reactions.	[36]
4	Interaction with Nucleic Acids	DNA Binding: LIG and its derivatives can interact with microbial DNA, causing structural changes or damage. This interaction can inhibit DNA replication and transcription, thereby preventing microbial proliferation. Genetic Damage: The oxidative stress induced by LIG can lead to mutations and breaks in the microbial DNA, further inhibiting cell viability.	[37]
5	Chelation of Metal Ions	Nutrient Deprivation: LIG can chelate essential metal ions such as iron and magnesium, which are necessary for microbial growth and enzyme function. By sequestering these ions, LIG deprives microorganisms of critical nutrients, inhibiting their growth. Metabolic Disruption: The chelation of metal ions can also disrupt various metabolic pathways that depend on these cofactors.	[38]
6	Chelation of Essential Nutrients	LIG can chelate metal ions like iron and magnesium, which are essential for microbial growth. By binding these ions, LIG deprives microorganisms of the nutrients they need for survival and proliferation.	[39]
7	Disruption of Electron Transport Chain	LIG can interfere with the electron transport chain in microbial cells, disrupting energy production and leading to cell death.	[40]
8	Binding to Proteins	LIG can bind to microbial proteins, causing structural changes or denaturation. This binding can inhibit enzyme activity and interfere with protein function, leading to microbial cell death.	[39]
9	Interference with Quorum Sensing	LIG can interfere with quorum sensing, the process by which bacteria communicate and coordinate their behavior. Disrupting quorum sensing can prevent biofilm formation and reduce virulence.	[41]
10	Modulation of Gene Expression	LIG can affect the expression of genes involved in microbial virulence and survival. This modulation can inhibit microbial growth and pathogenicity.	[42]
11	Surface Interaction and Biofilm Prevention	LIG can prevent the attachment of microorganisms to surfaces, thereby inhibiting biofilm formation. Biofilms protect microbes from external threats, so preventing their formation enhances antimicrobial efficacy.	[43]
12	Disruption of Membrane Potential	LIG can disrupt the membrane potential of microbial cells, affecting ion gradients and membrane permeability. This disruption can lead to cell death due to the inability to maintain essential cellular processes.	[44]
13	Interaction with Cell Membrane Proteins	LIG can bind to membrane proteins, altering their structure and function. This binding can inhibit nutrient transport, signal transduction, and other critical cellular functions.	[45]
14	Inhibition of ATP Synthesis	LIG can inhibit ATP synthesis in microbial cells by interfering with the function of ATP synthase or other components of the ATP production pathway, leading to energy depletion and cell death.	[46]
15	Induction of Apoptosis-like Cell Death	LIG can induce apoptosis-like cell death in microbial cells, characterized by cell shrinkage, DNA fragmentation, and other apoptotic markers. This programmed cell death can be triggered by oxidative stress and other cellular disruptions caused by LIG.	[47]
16	Activation of Antimicrobial Peptides	LIG can enhance the activity of naturally occurring antimicrobial peptides by interacting with them and increasing their affinity for microbial cell membranes.	[48].
17	Synergistic Effects with Other Antimicrobials	LIG can act synergistically with other antimicrobial agents, enhancing their efficacy. This synergy can occur through various mechanisms, such as disrupting microbial defenses or facilitating the entry of other antimicrobials.	[49]
18	Alteration of Microbial Metabolic Pathways	LIG can alter key metabolic pathways in microbial cells, leading to the accumulation of toxic intermediates or depletion of essential metabolites, which can inhibit growth and survival.	[50]

**Table 3 pharmaceuticals-17-01406-t003:** LIG antioxidant properties and modes of action (MOA).

#	Mechanism	Description	Ref.
1	Antioxidant Properties	LIG has significant antioxidant properties due to its phenolic structure, which allows it to scavenge free radicals. This activity is beneficial in reducing oxidative stress in biological systems and can be harnessed in the development of health supplements and pharmaceuticals.	[52]
2	Anti-inflammatory Effects	Some studies suggest that LIG and its derivatives can reduce inflammation by modulating the activity of inflammatory mediators. This potential makes LIG a candidate for developing treatments for inflammatory diseases.	[53]
3	Anticancer Potential	Certain LIG derivatives have been found to exhibit cytotoxic effects on cancer cells, inhibiting their growth and proliferation. This anticancer potential is an area of ongoing research, with the goal of developing LIG-based therapeutic agents.	[54]
4	Drug Delivery Systems	Due to its biocompatibility and biodegradability, LIG is being explored as a carrier material for drug delivery systems. LIG nanoparticles can encapsulate drugs, enhancing their stability and controlled release.	[55]
5	Prebiotic Activity	LIG can act as a prebiotic, promoting the growth of beneficial gut bacteria. This property is essential for maintaining a healthy digestive system and could be utilized in the development of functional foods and dietary supplements.	[56]
6	Environmental Applications	LIG-degrading microorganisms can be used to break down environmental pollutants, such as pesticides and industrial chemicals, making LIG a valuable tool in bioremediation efforts.	[57]
7	Bioremediation	LIG and its derivatives can enhance plant growth by improving soil structure, increasing nutrient availability, and protecting against pathogens. This can lead to more sustainable agricultural practices.	[58]

## Data Availability

Data sharing is not applicable.

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
