# Peer review of "Lignin: An Adaptable Biodegradable Polymer Used in Different Formulation Processes"

_pharmaceuticals, 2024, doi:10.3390/ph17101406_

Round 1
Reviewer 1 Report
Comments and Suggestions for Authors
Manuscript Number: pharmaceuticals-3253100
Title: Lignin: An Adaptable Molecule
Comments:
1. The title is too simple and does not highlight the main content of the article review.
2. The abstract has the same issue; although the author’s article structure is relatively clear, the abstract does not emphasize the main content and innovations of this review. Furthermore, there are already many reviews on lignin, so how does the author's review differ?
3. The entire article is lengthy and lacks relevant figures. The author should include more illustrations to engage readers. Additionally, a simple schematic diagram should be created in the introduction to provide a brief overview of the main structure of the article.
Author Response
Comment 1: The title is too simple and does not highlight the main content of the article review.
Response 1: The title was corrected as suggested: Lignine is an adaptable biodegradable polymer used in different formulation processes.
Comment 2: The abstract has the same issue; although the author’s article structure is relatively clear, the abstract does not emphasize the main content and innovations of this review. Furthermore, there are already many reviews on lignin, so how does the author's review differ?
Response 2: The abstract was corrected as suggested the following text was added: Lignine is a biopolymer found in vascular plant cell walls that is created by networks of hydroxylated and methoxylated phenylpropane that are randomly crosslinked. Plant cell walls contain lignin, a biopolymer with a significant potential for usage in modern industrial and pharmaceutical applications. It is a renewable raw resource. The plant is mechanically protected by this substance, which may increase its durability. Because it has antibacterial and antioxidant qualities, lignin also shields plants from biological and chemical challenges from the outside world. Researchers have done a great deal of work to create new materials and substances based on lignin. Numerous applications, including those involving antibacterial agents, antioxidant additives, UV protection agents, hydrogel-forming molecules, nanoparticles, and solid dosage forms, have been made with this biopolymer.
Comment 3: The entire article is lengthy and lacks relevant figures. The author should include more illustrations to engage readers. Additionally, a simple schematic diagram should be created in the introduction to provide a brief overview of the main structure of the article.
Response 3: More figures were added to the manuscript. Also, a brief diagram was added to the introduction part.
Figure 2 The structure of lignocellulosic biomass and its importance.(picture -personal collection)
Reviewer 2 Report
Comments and Suggestions for Authors
The document titled “” presents relevant information about lignin, extraction, purification, and several applications on different sectors such as drug-delivery, cosmetics, agronomy, pharmaceutic, among others. This information is the significant importance for the research community. However, the document needs substantial drafting improvement since some sections repeated information. In summary, the document requires a good proofreading and follows there are some extra recommendations
-Homogeneity in the text and abbreviations (LIG)
-Improve the drafting
-Improve the images, these are not complete clear
-Define all the abbreviations
-Add some additional examples of lignin extraction
-include a scheme of the obtaining process of vanillin
-include some extra images that represent the lignin applications
Comments on the Quality of English Languageno comments
Author Response
The document titled “” presents relevant information about lignin, extraction, purification, and several applications on different sectors such as drug-delivery, cosmetics, agronomy, pharmaceutic, among others. This information is the significant importance for the research community. However, the document needs substantial drafting improvement since some sections repeated information. In summary, the document requires a good proofreading and follows there are some extra recommendations
Comment1: -Homogeneity in the text and abbreviations (LIG)
Response 1: Lignin was replaced with LIG all over the text, and the first abbreviation was correctly cited. All abbreviations were rechecked.
Comment2: -Improve the drafting
Response2: drafting was improved
Comment 3: -Improve the images, these are not complete clear
Response 3: images were enhanced, and extra images were added as suggested.
Comment4: -Define all the abbreviations
Response 4: all abbreviations were checked and defined.
Comment5: -Add some additional examples of lignin extraction
Response5: corrected as suggested
Comment6: -include a scheme of the obtaining process of vanillin
Response6: correct as suggested.
Comment7: -include some extra images that represent the lignin applications
Response7: some extra images were added as sugested
Comment8: English language - no comments
Thank you for your review
Reviewer 3 Report
Comments and Suggestions for Authors
The manuscript titled “Lignin: An Adaptable Molecule” by Creteanu, A.; et al. is a Review work where the authors outlined the most recent advances in the exciting field of lignin as renewable biopolymer to minimize the use of scarce fossil fuel resources. A complete overview is depicted in this manuscript being a topic of growing interest. Furthermore, the manuscript is generally well-written. However, it exists some points that need to be addressed (please, see them below detailed point-by-point) to improve the scientific quality of the submitted manuscript paper before this article will be consider for its publication in Pharmaceuticals.
1) The authors should consider to add the terms “biosynthesis” and “antimicrobial properties” in the keyword list.
2) “Lignin, a complex organic polymer, is a critical player in plant biology (…) Moreover, as a renewable resource, lignin is being explored for conversion into biofuels, chemicals, and materials, contributing to the development of sustainable biorefineries” (page 1). Could the authors provide quantitative data insights according the worldwide economic impact of lignin in the Industrial sectors mentioned in this statement? This will significantly aid the potential readers to better understand the significance of this Review work.
3) Then, in many statements “Lignin” terms appears in capital letter and it should be exchanged by lowercase lettering. This point should be taken into account for the entire manuscript body text.
4) “1.2. Lignin synthesis” (pages 3-4). Some information concerning the reaction yields should be also furnished in this subsection.
5) “4. Research and application” (pages 7-26). First, some additional figure illustrations the potential applications of lignin biobased materials should be furnished. This figures could come to recently published works. Then, it is also necessary to mention the excellent moisture barrier performance of lignin independently of the environmental relative humidity [1] and how lignin can be exploited for food-packaging materials [2].
[1] Marcuello, C.; Foulon, L.; Chabbert, B.; Aguié-Béghin, V.; Molinari, M. Atomic force microscopy reveals how relative humidity impacts the Young’s modulus of lignocellulosic polymers and their adhesion with cellulose nanocrystals at the nanoscale. Int. J. Biol. Macromol. 2020, 147, 1064-1075. https://doi.org/10.1016/j.ijbiomac.2019.10.074
[2] Javed, A.; et al. Lignin-Containing Coatings for Packaging Materials-Pilot Trials. Polymers 2021, 13, 1595. https://doi.org/10.3390/polym13101595
6) Conclusions (pages 26-27). This section perfectly remarks the most relevant outcomes found by the authors in this field and the promising future perspectives. It should be desirable to add a brief statement to discuss about the potential future action lines to pursue the topic covered in this research. Finally, the grammar of this sentence needs to be revisited and corrected “This is a very imortanta parameter for the formulaion of (…)”.
Comments on the Quality of English LanguageThe manuscript is generally well-written albeit it may be desirable if the authors could recheck it in order to polish those final details susceptible to be improved.
Author Response
The manuscript titled “Lignin: An Adaptable Molecule” by Creteanu, A.; et al. is a Review work where the authors outlined the most recent advances in the exciting field of lignin as renewable biopolymer to minimize the use of scarce fossil fuel resources. A complete overview is depicted in this manuscript being a topic of growing interest. Furthermore, the manuscript is generally well-written. However, it exists some points that need to be addressed (please, see them below detailed point-by-point) to improve the scientific quality of the submitted manuscript paper before this article will be consider for its publication in Pharmaceuticals.
Comment 1: 1) The authors should consider adding the terms “biosynthesis” and “antimicrobial properties” to the keyword list.
Response1: corrected as suggested.
Comment2: 2) “Lignin, a complex organic polymer, is a critical player in plant biology (…) Moreover, as a renewable resource, lignin is being explored for conversion into biofuels, chemicals, and materials, contributing to the development of sustainable biorefineries” (page 1). Could the authors provide quantitative data insights according the worldwide economic impact of lignin in the Industrial sectors mentioned in this statement? This will significantly aid the potential readers to better understand the significance of this Review work.
Response2: corrected as suggested. Thje following text was added: The global lignin market was valued at USD 1.08 billion in 2023 and is projected to expand at a compound annual growth rate (CAGR) of 4.5% from 2024 to 2030. The rising demand for lignin in animal feed and natural goods is expected to stimulate growth. The product is extensively employed in the synthesis of macromolecules for the manufacturing of bitumen, biofuels, and biorefinery catalysts. This aspect is expected to facilitate market expansion. The COVID-19 pandemic adversely affected the market. This occurred due to the closure of manufacturing facilities and plants as a result of the lockdown and limitations. Disruptions in supply chain and transportation exacerbated obstacles for the sector. The industry encountered a backlash due to disturbances in the value chain, encompassing staff reductions, raw material shortages, trade and transportation challenges, and unpredictable customer demand..
Comment 3: 3) Then, in many statements “Lignin” terms appears in capital letter and it should be exchanged by lowercase lettering. This point should be taken into account for the entire manuscript body text.
Response 3: lignin was replaced with the abbreviation LIG all over the text.
Comment 4: 4) “1.2. Lignin synthesis” (pages 3-4). Some information concerning the reaction yields should be also furnished in this subsection.
Response 4: lingnin sintesis section was expended with a comprehensive table as suggested(Table 1) .
Comment5: 5) “4. Research and application” (pages 7-26). First, some additional figure illustrations the potential applications of lignin biobased materials should be furnished. This figures could come to recently published works. Then, it is also necessary to mention the excellent moisture barrier performance of lignin independently of the environmental relative humidity [1] and how lignin can be exploited for food-packaging materials [2].
[1] Marcuello, C.; Foulon, L.; Chabbert, B.; Aguié-Béghin, V.; Molinari, M. Atomic force microscopy reveals how relative humidity impacts the Young’s modulus of lignocellulosic polymers and their adhesion with cellulose nanocrystals at the nanoscale. Int. J. Biol. Macromol. 2020, 147, 1064-1075. https://doi.org/10.1016/j.ijbiomac.2019.10.074
[2] Javed, A.; et al. Lignin-Containing Coatings for Packaging Materials-Pilot Trials. Polymers 2021, 13, 1595. https://doi.org/10.3390/polym13101595
Response5: Corrected as suggested.
Comment6: 6) Conclusions (pages 26-27). This section perfectly remarks the most relevant outcomes found by the authors in this field and the promising future perspectives. It should be desirable to add a brief statement to discuss about the potential future action lines to pursue the topic covered in this research. Finally, the grammar of this sentence needs to be revisited and corrected “This is a very imortanta parameter for the formulaion of (…)”.
Response6: Conclusions have been improved
Comment7 English language: The manuscript is generally well-written, albeit it may be desirable if the authors could recheck it in order to polish those final details susceptible to be improved.
Response7: English was corrected..